# OmniDraft: A cross-vocabulary, online adaptive drafter for on-device speculative decoding

**Ramchalam Kinattinkara Ramakrishnan**[*, 1] **Zhaocong Yuan**[*, 2],
Shaojie Zhuo[3], Chen Feng[4], Yicheng Lin[5], Chenzheng Su[6], Xiaopeng Zhang[7]

Qualcomm AI Research [†]
{[1]rkinatti, [2]zhaocong, [3]shaojiez, [4]chenf, [5]yichengl, [6]chenzhen, [7]xiaopeng}
@qti.qualcomm.com

## Abstract

Speculative decoding generally dictates having a small, efficient draft model that is either pretrained or distilled offline to a particular target model series, for instance, Llama or Qwen models. However, within online deployment settings, there are two major challenges: 1) usage of a target model that is incompatible with the draft model; 2) expectation of latency improvements over usage and time. In this work, we propose OmniDraft, a unified framework that enables a single draft model to operate with any target model and adapt dynamically to user data. We introduce an online n-gram cache with hybrid distillation fine-tuning to address the cross-vocabulary mismatch across draft and target models; and further improve decoding speed by leveraging adaptive drafting techniques. OmniDraft is particularly suitable for on-device LLM applications where model cost, efficiency and user customization are the major points of contention. This further highlights the need to tackle the above challenges and motivates the *"one drafter for all"* paradigm. We showcase the proficiency of the OmniDraft framework by performing online learning on math reasoning, coding and text generation tasks. Notably, OmniDraft enables a single Llama-68M model to pair with various target models including Vicuna-7B, Qwen2-7B and Llama3-8B models for speculative decoding; and additionally provides up to 1.5-2x speedup.

## 1 Introduction

Unlike traditional auto-regressive generation in LLMs, speculative decoding (SpD) [25, 9] offers a unique advantage to accelerate LLM inference by decoupling the generation phase and verification phase. Speculative decoding generally requires a small but efficient draft model and a large target model. The draft model generates a sequence of proposed tokens to be verified by the target in one shot, amortizing the target model's memory bottleneck in batch inference and attaining better tokens per second throughput. The speedup factor relies not only on the predictability of the generated text like commonly occurring phrases, but also on the alignment between draft and target model. As such, a common practice is to utilize draft and target models from the same model family given their consistency in pretrained data, tokenization and training configurations. Alternatively, one might consider distilling a target model into a smaller model to serve as the drafter [53, 30], which still follows the same principle of better alignment leading to greater speedup.

The tight coupling of draft and target models limits flexibility of model selection and creates additional overhead for draft model distillation and maintenance, especially when deploying LLMs at scale

---

[*]Contributed equally

[†]Qualcomm AI Research is an initiative of Qualcomm Technologies, Inc.

39th Conference on Neural Information Processing Systems (NeurIPS 2025).

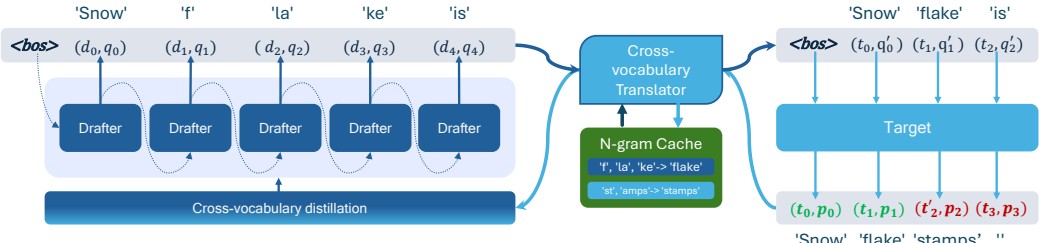

Figure 1: Overview of the OmniDraft framework: during cross-vocabulary speculative decoding, the drafter (Llama-68M) generates multiple tokens $d_i$ with corresponding distributions $q_i$. Cross-vocabulary translator then converts the drafter tokens into tokens in the vocabulary of the target model (Llama3-8B). In this example, token $d_0$('Snow') and $d_4$('is') are directly mapped to target tokens $t_0$ and $t_2$, while token $d_1$('f'), $d_2$('la') and $d_3$('ke') are merged into a single target token $t_1$ ('flake'), since there is a mapping item in the n-gram cache. The translated proposal $t_i$ along with combined probabilities $q_i'$ is verified by the target model, resulting in $t_0$ and $t_1$ being accepted while $t_2$ being rejected and replaced by $t_2'$. The target outputs tokens and their probabilities $p_i$ are translated into drafter tokens and sent back to drafter for next round of drafting. The n-gram cache is updated by inserting a new unseen item ('st','amps'->'stamps'). Meanwhile, the accepted and corrected tokens from the target model are used to align the drafter through online cross-vocabulary distillation.

across diverse hardware platforms and updating target models overtime. It is compelling to use a universal lightweight model on-device to draft tokens for a broad range of targets models. This universality greatly simplifies deployment, facilitates easy optimization, and allows for rapid model updates. Furthermore, the heterogeneity of on-device and cloud hardware presents an opportunity for hybrid speculative decoding. This enables users to choose between running any target model locally or on the cloud, balancing model performance, inference cost, and privacy concerns.

However, building a universal drafter for speculative decoding presents several unique challenges. Firstly, different family of target models might use tokenizers with different vocabularies. This is natural since target models are typically trained with massive pretrained data and hence a larger vocabulary is needed to include higher-order n-grams or BPE [7] merges. As a result, the vocabulary mismatch breaks the speculative decoding formulation where the draft and target model need to evaluate probabilities over the same set of tokens. A previous work UAG [45] addresses vocabulary mismatch with an intermediate translation layer, but it primarily deals with tokens within the intersection of the drafter and target vocabularies, which might "falsely" reject good tokens from the drafter. Secondly, independent training of the drafter and target models can result in misaligned predicted token distributions, which reduces the acceptance rate during verification. This misalignment diminishes the efficiency benefits of speculative decoding. Although existing research [53, 30, 34] has explored techniques to improve alignment between the drafter and target, the alignment is often done offline with assumption of a fixed target model. In practice, the target model may change due to user personalization or tasks switching, further complicating the alignment issue. Additionally, edge devices usually have limited memory, compute capacity, and power budget. The drafting process must therefore be efficient to maximize the benefits of speculative decoding.

To address the challenges, we propose a scalable speculative decoding framework, named *OmniDraft*, centered on an on-device universal drafter that generates draft tokens for a wide variety of target models. An overview of *OmniDraft* is shown in Figure 1. To tackle the vocabulary mismatch issue and enable cross-vocabulary speculative decoding, we introduce an n-gram cache to store cross-vocabulary mappings from draft tokens to target tokens. By integrating the n-gram cache into the speculative decoding algorithm, we alleviate vocabulary mismatches and achieve a higher acceptance rate for future queries. We further employ online knowledge distillation to improve alignment between the drafter and target. A hybrid distillation loss, combining token-level and distribution level objectives, updates the drafter using the target's accepted and corrected outputs. This enables continuous alignment during speculative decoding, even when the target model changes due to personalization or task switching. To further improve runtime efficiency, we incorporate online adaptive drafting, where the drafter dynamically adjust the number of tokens it proposes

based on predicted confidence. The adaptive drafting balances generation cost and acceptation rate, maximizing throughput under device constraints.

Overall, our OmniDraft framework enables a robust, efficient and flexible speculative decoding system with a universal drafter for on-device applications. Through extensive experiments, we show that a single Llama-68M draft model can be paired with various target models including Vicuna-7B, Qwen2-7B and Llama3-8B models for cross-vocabulary speculative decoding and provides up to 1.5-2x speedup on reasoning, coding and text generation tasks.

**Contributions** (1) We propose cross-vocabulary speculative decoding via online n-gram cache that translates between drafter and target vocabularies, enabling speculative decoding across models with different tokenizers; (2) We introduce online knowledge distillation with hybrid alignment loss, which updates the drafter using accepted and corrected outputs from the target model to improve alignment and acceptance rate over time; (3) We integrate alignment training with adaptive drafting, allowing the drafter to dynamically adjust draft length based on alignment confidence for improved efficiency and speedup.

## 2    Related work

**Speculative decoding**    The idea of speculative decoding is proposed and formalized in the pioneer works of [25, 9]. Subsequent works have centered around optimizing different components of the speculative decoding framework. Some have highlighted tree attention to facilitate simultaneous verification of multiple draft sequences such as SpecInfer [32], Medusa [8] and Sequoia [11]. There is also focus on more efficient drafting by using retrieval-based methods as in Lookahead [16], REST [19], NEST [26], RASD [38], or by using dynamic length drafting as in BiLD [22], DISCO [31], AdaEDL [2], SpecDec++ [21], EAGLE2 [28].

More recent works explore speculative decoding in the context of vocabulary adaptation like UAG [45] and AdaptiVocab [35], long-context tasks like LongSpec [50], MagicDec [41], or more efficient draft models as in EAGLE [27], Speculative Streaming [5], and self speculative decoding as in [51], [14], [48].

**Distillaton**    Model distillation in LLMs is crucial for speculative decoding since quality of the draft model dictates the final speed-up. Early works on sequence model distillation include [3, 23, 46]. There are also recent works that have specific focus on LLM distillation such as MiniLLM [18] and GKD [1]. Most closely aligned to our setting are the works of DistillSpec [53] and OSD [30] that aim towards training a better draft model to a given target. Another related line of works is distillation on different tokenizers as in [6, 33, 34].

**Online adaptation**    Compared to model finetuning on a fixed offline training set over multiple epochs, online adaptation focuses on continual learning or few-shot generalization to new data. Frameworks such as ProtoNet [42] and MAML [15] address the few-shot learning problem. Robotics and reinforcement learning also offer insights with works like DAGGER [40], RL^2 [13] and PEARL [39] for continuous adaptation to new tasks. There are also works that emphasize online adaptation for LLMs as in [20, 43, 29]. Lastly in speculative decoding, OSD [30] explicitly optimizes for draft model online adaptation which is closest to ours.

## 3    Methodology

**Notation**    Assume a small draft model $M_q$ and a large target model $M_p$, let $p(y_t|x, y_{<t})$ and $q(y_t|x, y_{<t})$ be the distributions of next-token predictions at time step $t$ for $M_p$ and $M_q$ respectively, where $x$ represents the prompt prefix and $x, y_{<t}$ represents the context at time step $t$. For convenience, we use $p(y_t)$ and $q(y_t)$ as shorthands for $p(y_t|x, y_{<t})$ and $q(y_t|x, y_{<t})$ for the remaining sections. $k$ denotes the number of tokens proposed by the drafter $M_q$ and $p'(y_{t+i}) = \mathrm{norm}(\max(0, p(y_{t+i}) - q(y_{t+i})))$ is the residual distribution for the resampling, as per the original SpD algorithm [25, 9]

[53, 30, 1] shows that better alignment between the draft and target model gives higher acceptance rate and hence higher speedup in speculative decoding. To align them, some common distillation losses include supervised finetuning (FT) or sequence level Knowledge Distillation (KD) $\mathcal{L}_{SFT}(\theta) =$

$\mathbb{E}_{(x,y)\sim(X,Y)}[-\log q_\theta(y|x)]$, and supervised KD $\mathcal{L}_{SD}(\theta) = \mathbb{E}_{(x,y)\sim(X,Y)}[\mathcal{D}(M_p||M_q^\theta)(y|x)]$ with a selected choice of divergence metric $\mathcal{D}$.

## 3.1 Cross-vocabulary N-gram Cache

Normal speculative decoding is infeasible when the draft vocabulary $V_q$ and target vocabulary $V_p$ are different, since the rejection scheme relies on acceptance ratios evaluated on the same token $\min(1, p(y_{t+i})/q(y_{t+i}))$ and the residual distribution $\mathrm{norm}(\max(0, p(y_{t+i})-q(y_{t+i})))$ also requires per-token probability differences. This foreshadows two issues: 1) tokens without direct mapping between the drafter vocabulary and target vocabulary cannot be handled i.e. the drafter can propose tokens not recognized by the target or vice-versa. 2) target can "falsely" reject good tokens from the drafter. This occurs when the drafter proposes a sequence of (sub-)tokens that constitute a merged token/n-gram in target, but target rejects the sequence since it prioritizes the merged token over the prefix sub-token. This is a byproduct of the tokenization process that optimizes for the longest token in the vocabulary or sequentially applies merge rules to get the longest possible token ([47, 24, 7]).

UAG [45] addresses the first mismatch with a translation layer between the drafter and target and derives the intersection of draft and target vocabularies, where tokens have direct mappings. During proposal stage, UAG suppresses tokens outside of the intersection and converts drafter token ids to target ids with the mapping. After verification, if a token without direct mapping is sampled, the translation layer will invoke the target and draft tokenzier to map the target token to sub-tokens in draft vocabulary. However, UAG cannot solve the second mismatch meaning it only guarantees feasibility of cross-vocabulary speculative decoding but lacks in optimality.

To overcome this, we propose to build a cache of n-grams $\mathcal{C}$ that tracks the instances of target-draft token translations. Denote target tokens $t_i \in V_p$ and drafter tokens $d_i \in V_q$, then the n-grams cache is $\mathcal{C} = \{(t_i, [d_j^i]_{j=1:n})\}$, where each element represents a mapped n-gram instance given the matching context so far $ctx_q, d_1^i, d_2^i, \cdots, d_n^i = \mathrm{tokenize}_q(\mathrm{detokenize}_p(ctx_p, t_i))$. In inference time, we add a postprocessing (pp) stage at the translation layer, where we scan over the proposed draft tokens $d_t, \cdots, d_{t+k-1}$ and merge sub-tokens that hit the n-gram cache. The resulting new sequence $t_t, \cdots, t_{t+m}$ and their draft probabilities $q'(t_t), \cdots, q'(t_{t+m})$ will be under the target vocabulary space and follow the mapping rule

$$t_i, q'(t_i) = \begin{cases} d_i, q(d_i) & \text{if direct mapping, } t_i \leftrightarrow d_i \\ \mathrm{lookup}([d_j^i]_{j=1:n}, \mathcal{C}), \prod_j q(d_j^i) & \text{otherwise} \end{cases} \tag{1}$$

This cross-vocabulary mapping translates the draft sequence to the target vocabulary space, ensuring speculative decoding still functions well with per-token acceptance ratios $\min(1, p(t_{t+i})/q'(t_{t+i}))$. From the perspective of the final matched text, $p(t_i)$ and $\prod_j q(d_j^i)$ would be the probability of producing that specific chunk of text in their respective tokenization space.

However, in the correction stage we require the full distribution for the residual distribution instead of point-wise evaluation of probabilities, which is infeasible since $p(\cdot), q(\cdot)$ work on different space. Hence we enhance the mapping rule 1 as

$$\forall t \in V_p, \quad q'(t) = \begin{cases} \prod_j q(d_j^i) & \text{if n-gram mapped, } t \leftrightarrow [d_j^i]_{j=1:n} \\ q(d_1^i) - \prod_j q(d_j^i) & \text{prefix sub-token of n-gram, } t = d_1^i \\ q(t) & \text{otherwise} \end{cases} \tag{2}$$

We use the mapped probability for the selected n-gram but adjust the prefix sub-token probability by subtracting the n-gram probability. This can be seen as approximately re-allocating the original probability mass assigned to prefix sub-token $d_1^i$ under the draft distribution $q(\cdot)$, between the "new" n-gram token and the prefix sub-token under the modified draft distribution $q'(\cdot)$. This is also related to the known problem of tokenization bias [37]. Using mapping 2, we at least ensure point-wise, approximate correctness of the n-gram token for the residual distribution $\mathrm{norm}(\max(0, p(t_{t+i}) - q'(t_{t+i})))$. Note that we can apply 2 to the n-grams sampled and matched from the current speculative round. Evaluating all other n-grams require re-running drafter at the step of rejection which is impractical. We summarize the modified speculative decoding with our proposed mappings in Algorithm 1.

---

**Algorithm 1** Cross-vocabulary Speculative Decoding

---

1: Given draft model $q(\cdot)$, target model $p(\cdot)$, n-gram cache $\mathcal{C}$
2: Given draft length $k$, max length $T$, prompt $x$
3: Initialize $t \leftarrow 0, ctx_q, ctx_p \leftarrow x$
4: **while** t < T **do**
5:     **for** $i = 1 : k$ **do**
6:         Sample draft auto-regressively $d_{t+i} \sim q(d_{t+i}|ctx_q, d_{<t+i})$
7:     **end for**
8:     Apply translation mapping 12 to get $m$ proposed tokens and draft probabilities in target space
9:            $t_t, \cdots, t_{t+m-1}, q'(t_t), \cdots, q'(t_{t+m-1})$
10:    In parallel, compute target probabilities on mapped tokens
11:         $p(t_t), \cdots, p(t_{t+m})$
12:    Apply rejection sampling with acceptance ratios $\min(1, p(t_{t+i})/q'(t_{t+i}))$ and correction residual distributions $\mathrm{norm}(\max(0, p(t_{t+i}) - q'(t_{t+i})))$ to get $n$ accepted tokens
13:         $t_t, \cdots, t_{t+n-1}$ for some accepted length $n$
14:    **if** $n == m$ **then**
15:        Sample free token $t_{t+m} \sim p(t_{t+m})$ and add to accepted tokens, $n \leftarrow m + 1$
16:    **end if**
17:    Apply reverse translation to get accepted $p$ tokens in draft space
18:         $ctx_q, d_t, \cdots, d_{t+p-1} = \mathrm{tokenize}_q(\mathrm{detokenize}_p(ctx_p, t_t, \cdots, t_{t+n-1}))$
19:    Add to n-gram cache $\mathcal{C}$ if there exists unseen n-gram instance
20:    $t \leftarrow t + n$, update $ctx_q, ctx_p$
21: **end while**
22: Return results

---

## 3.2 Cross-vocabulary Distillation

Using the n-gram cache with the approximate distribution mapping helps to draft and verify n-gram tokens as if operating under the target vocabulary directly. To extend it for online adaptation, we propose a hybrid distillation framework that progressively aligns the draft and target model on both direct mapping tokens and n-gram tokens. Given the online setting, we have limited access to the target model so we distill on the draft model generated data, or simply on-policy data similar to GKD [1]. We employ reverse KL on direct mapping tokens for richer supervision signals, but use maximum log-likelihood (NLL) on n-gram tokens since we only have reliable point-wise evaluation of probabilities on those tokens. Overall, our proposed hybrid distillation loss is

$$\mathcal{L}_{\text{cross\_vocab\_distill}}(\theta) = \mathcal{L}_{\text{DM}}(\theta) + \lambda \mathcal{L}_{\text{N-gram}}(\theta) \tag{3}$$

$$= \mathop{\mathbb{E}}_{\substack{x \sim X, d_i \sim q(\cdot), \\ t_i, q' \leftarrow \text{mapping}(d_i, q)}} \left[ \mathcal{D}_{KL}(q'_\theta || p)(t_i|x)\mathcal{I}_{\text{DM}}(d_i) - \lambda \log q_\theta(d_i|x)\mathcal{I}_{\text{N-gram}}(d_i) \right] \tag{4}$$

where $\mathcal{I}_{\text{DM}}, \mathcal{I}_{\text{N-gram}}$ are the indicator functions to identify if current token is part of direct mapping or n-gram; this is implemented as binary masks in practice. Note that the KL loss term corresponds to direct mapping token $t_i \leftrightarrow d_i$, and divergence is computed in the target vocabulary space given $q(\cdot)$ is elevated to $q'(\cdot)$ via the translation mapping. The NLL loss term however operates in drafter vocabulary space to increase likelihoods of drafter tokens which constitute an n-gram accepted by the target during inference.

The parameter $\lambda$ can either be a hyperparameter to account for ratio imbalance between direct mapping tokens and n-gram tokens, or can be a dynamic weight such as the verified target probability of the n-gram. The latter leads to a loss term of $\lambda \mathcal{L}_{\text{N-gram}}(\theta) = -p(t_i) \log q_\theta(d_i|x)$, which can be treated as the point-wise KL evaluated on the n-gram token.

Moreover, it is possible to extend the NLL or point-wise KL loss to an approximate KL loss using mapping 2 as $\mathcal{L}_{\text{N-gram}} = \mathcal{D}_{KL}(q'_\theta(t_i|x) || p(t_i|x))$. This is equivalent to using KL on the intersection tokens plus the n-gram and it's first sub-token. Due to the additional components on the intersection tokens and the fist sub-token, this approximate KL loss can provide richer learning signal. Empirically we only observed minimal improvement from the approximate KL loss, as shown in section 4.3.4.

## 3.3 Online Adaptive Drafting

One observation in performing cross-vocabulary speculative decoding is that we are implicitly shortening the proposal draft length, since multiple sub-tokens map to a single n-gram token. We then explicitly incorporate adaptive drafting to gain even better speedup for on-device speculative decoding focusing on the same vocab setting. We adopt the framework in SpecDec++ [21] where a lightweight head network $f_\phi(\cdot)$ predicts the acceptance rate of the current proposed token. The acceptance prediction head takes the embedding of the proposed token $e_i$ as input, and is trained using weighted BCE loss $\mathcal{L}_{\text{adapt}}$ with acceptance ratios $\min(1, p(y_i)/q(y_i))$ as labels. It then controls if to early exist based on the cumulative probability of at least one proposed token getting rejected and a given stopping threshold $\gamma$.

$$P(y_i \text{ accepted}|y_{<i} \text{ accepted}) = \text{sigmoid}(f_\phi(e_i)) \tag{5}$$

$$P(\exists 1 \le i \le k, \ s.t. \ y_i \text{ rejected}) = 1 - \prod_{i=1}^{k} P(y_i \text{ accepted}|y_{<i} \text{ accepted}) \tag{6}$$

$$P(\exists 1 \le i \le k, \ s.t. \ y_i \text{ rejected}) > \gamma \implies \text{exit} \tag{7}$$

However in online adaptation, the labels are subject to change since the draft model is continuously finetuned with distillation loss $\mathcal{L}_{\text{distill}}$ to align with the target, which could cause distribution shift for dynamic drafting.

We propose two variants of online adaptive drafting. The first one performs draft model alignment and acceptance prediction head training jointly at each update step $\mathcal{L}_{\text{joint}} = \mathcal{L}_{\text{distill}} + \mathcal{L}_{\text{adapt}}$. The second one interleaves two trainings such that we perform multiple acceptance prediction updates per draft model alignment update, aiming to mimic slowly moving labels to reduce distribution shift. Unlike the first variant that performs both updates jointly using the online data batch, for the second variant we keep a larger buffer for the acceptance prediction updates which includes data from previous batches. This helps to enhance training stability and adaptation speed for the acceptance prediction.

# 4 Results

**Models**   To show the efficacy of OmniDraft on the setting of a single drafter for multiple targets, we fix the drafter to be Llama-68M [32] and the target model to be Llama3-8B [17], Qwen2-7B [49] for the cross vocabulary results, as well as Vicuna-7B [52] for the same family vocabulary evaluation.

**Tasks**   We perform online distillation across 4 tasks: GSM8K [12], Alpaca [44], XSum [36]and a combined MBPP+HumanEval [4][10] datasets. Each task has a dedicated train and test set or we slice out the a portion of the train set as the test set. For the MBPP+HumanEval, we combine the two datasets to add some more diversity to the data for the coding tasks. The training is conducted for a specific number of steps (<1 epoch) across each of the tasks as per general online adaptation setting. All the tokens will contribute to the loss calculation including the tokens that were accepted by the target as they would provide option for improved alignment between the two models. Moreover, all experiments are performed with temperature 0.01, unless specified otherwise. We also include the setting for online adaptation of the drafter using LoRA across all the tasks. Using dynamic adapter switching we can pair the same drafter with any target across any of the task which is the ideal scenario for on-device online speculative decoding.

**Evaluation Metrics**

- **Speedup** - Walltime acceleration rate, measures the improvement in tokens-per-second throughput. We follow Medusa's [8] convention.
- **Acceptance Rate** - Ratio of accepted tokens to proposed tokens averaged over speculative decoding steps, measures alignment between draft and target model.

**Notation**   We refer to $\text{SpD}_{DM}$ as the baseline speculative decoding which uses direct mapping between vocabularies. N-gram postprocessing (pp) refers to using the N-gram cache as a postprocessing technique without directly training on it as we proposed in Algorithm 1. N-gram hit refers to the average number of successful N-gram cache lookups that were accepted by the target per speculative decoding step.

## 4.1 Cross-vocabulary Online Distillation

Table 1 shows the results on the test set after training. Across all the tasks, $\mathcal{L}_{\text{DM}} + \mathcal{L}_{\text{N-gram}}$ approach perform better than training only on $\mathcal{L}_{\text{DM}}$. Overall, this indicates that the additional $\mathcal{L}_{\text{N-gram}}$ plays a significant role in improving the acceptance rate. Moreover, $\mathcal{L}_{\text{DM}} + \mathcal{L}_{\text{N-gram}}$ with LoRA finetuning performs reasonably well across all of the tasks when compared to the baseline. The largest speedup is obtained for the GSM8K dataset for both the target models, with XSum being the least improved task. We observe XSum could also be improved by increasing the number of training samples as a scaling effect. Figure 2 portrays the training dynamics across all tasks for both the acceptance rate and speedup metrics. For all the experiments, we can see the metrics improve as training proceeds. The instability seen in some of the LoRA curves could indicate that training hyper-parameters are not fully optimized or that training plateaus quicker for certain tasks, which could also explain why there is still a small gap between the LoRA and the full finetuned model performance.

Table 1: Performance on Cross-vocabulary Distillation with Llama-68M and two different targets

| Target | Method | GSM8K | | MBPP+HumanEval | | Alpaca | | XSum | |
|---|---|---|---|---|---|---|---|---|---|
| | | Acc Rate | Speedup | Acc Rate | Speedup | Acc Rate | Speedup | Acc Rate | Speedup |
| Llama3-8B | $\text{SpD}_{DM}$ | 0.10 | 0.94x | 0.09 | 1.03x | 0.09 | 0.96x | 0.11 | 0.91x |
| | $\mathcal{L}_{\text{DM}}$ | 0.32 | 1.58x | 0.22 | 1.26x | 0.16 | 1.25x | 0.20 | 1.20x |
| | $\mathcal{L}_{\text{DM}} + \lambda\mathcal{L}_{\text{N-gram}}$ | **0.42** | **1.70x** | **0.27** | **1.33x** | **0.20** | **1.30x** | **0.24** | **1.24x** |
| | $\mathcal{L}_{\text{DM}} + \lambda\mathcal{L}_{\text{N-gram}} + \text{LoRA}$ | 0.37 | 1.59x | 0.19 | 1.28x | 0.17 | 1.21x | 0.23 | 1.21x |
| Qwen2-7B | $\text{SpD}_{DM}$ | 0.14 | 1.04x | 0.09 | 0.91x | 0.13 | 1.01x | 0.12 | 0.96x |
| | $\mathcal{L}_{\text{DM}}$ | 0.33 | 1.50x | 0.22 | 1.29x | 0.17 | 1.25x | 0.19 | 1.16x |
| | $\mathcal{L}_{\text{DM}} + \lambda\mathcal{L}_{\text{N-gram}}$ | **0.37** | **1.61x** | **0.26** | **1.36x** | **0.20** | **1.30x** | **0.22** | **1.22x** |
| | $\mathcal{L}_{\text{DM}} + \lambda\mathcal{L}_{\text{N-gram}} + \text{LoRA}$ | 0.31 | 1.41x | 0.21 | 1.21x | 0.18 | 1.25x | 0.22 | 1.21x |

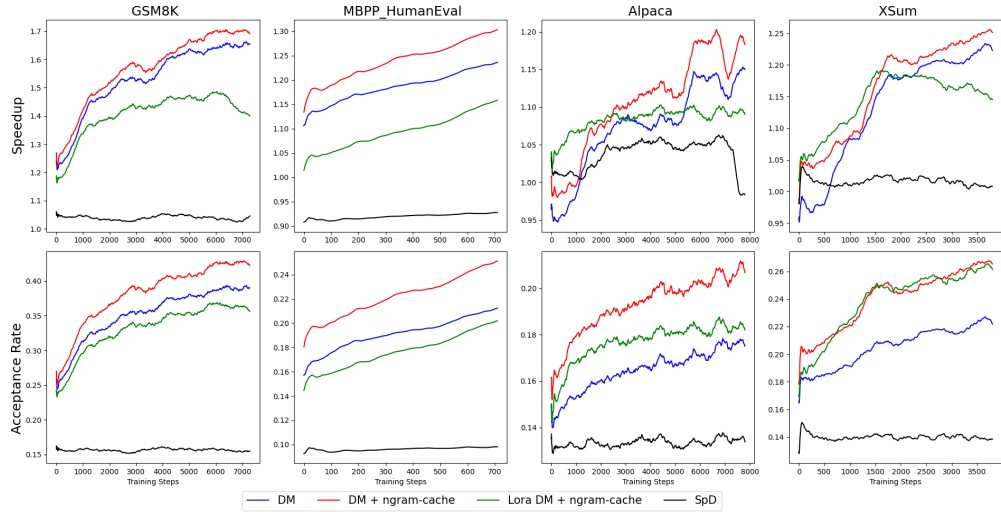

Figure 2: Cross-vocabulary SpD online distillation on Llama-68M with Qwen2-7B as target

## 4.2 Online Adaptive Drafting

Table 2 shows the test results of online adaptive drafting with ablations to the different training variants. Across all tasks we observe a consistent increase in acceptance rate. This is the combined effect of better model alignment due to distillation, and also the acceptance prediction head learning to exit early when there is a higher chance for proposal rejection

In terms of speedup, we can also see improvement in most tasks except on GSM8K where the distill-only baseline outperforms both adaptive drafting variants. This could be due to the latency reduction over the number of the proposed tokens are not sufficient in comparison to the increase in successful accepted tokens with longer draft length. We hypothesize it could be due the difficulty in training the acceptance prediction head in an online setting. This involves optimizing with respect to

changing labels and training only on incoming data without a stable corrective feedback compared to offline training. By end of the one epoch training, the acceptance prediction head might not have converged well to align with the current distilled drafter, leading to sub-optimal performance.

Finally, we also observe the interleaved variant has better speedup than the joint variant on average (similar in Alpaca and Xsum, larger increment in GSM8K and MBPP+HumanEval). But the joint variant has higher acceptance rates over all tasks, indicating it might be underestimating the acceptance probability, leading to wrongful early exit. This could be a direct consequence of the previously mentioned training challenges, which is alleviated with the interleaved variant with more stable training.

Table 2: Performance on Online Adaptive Drafting with Llama-68M and Vicuna-7B.

| Target | Method | GSM8K | | MBPP+HumanEval | | Alpaca | | XSum | |
|---|---|---|---|---|---|---|---|---|---|
| | | Acc Rate | Speedup | Acc Rate | Speedup | Acc Rate | Speedup | Acc Rate | Speedup |
| Vicuna-7B | SpD | 0.21 | 1.44x | 0.14 | 1.22x | 0.20 | 1.44x | 0.20 | 1.42x |
| | Adapt Only | 0.38 | 1.50x | 0.28 | 1.34x | 0.38 | 1.52x | 0.38 | 1.51x |
| | Distill Only | 0.42 | **2.20x** | 0.35 | 1.92x | 0.25 | 1.57x | 0.23 | 1.53x |
| | Joint Distill + Adapt | **0.61** | 2.08x | **0.51** | 1.91x | **0.44** | **1.61x** | **0.42** | **1.59x** |
| | Interleaved Distill + Adapt | 0.52 | 2.15x | 0.48 | **1.94x** | 0.41 | 1.60x | 0.38 | 1.58x |

## 4.3 Ablations Studies

### 4.3.1 Scalability to Larger Target and Draft Models

We perform an ablation on scaling the target and draft model and compare the various metrics and whether it follows a similar trend. As shown in Table 3, we used a new family of LLMs, the Qwen2.5 series of models of larger sizes including 7B, 14B and 32B parameters. Datasets used are GSM8k and MBPP+HumanEval. We also performed analysis on a larger drafter of size 160M parameters. The performance on these larger models are consistent with our previous results, with larger target models resulting in 2x improvement on the GSM8K dataset. The Omnidraft framework demonstrates strong scalability as the size of the target LLM increases, as shown in the table. While the gap between the drafter and the target model grows with model size, the drafter remains the limiting factor. During training, the drafter progressively aligns with the target; however, once it reaches its alignment capacity, further increase in target size continues to yield speedup improvements, while the acceptance rate plateaus. Regarding the larger drafter, the Llama-160M parameter model was chosen after taking into consideration the increased latency of around 3x of the drafter model compared to Llama-68M. Overall, the larger drafter model does introduce a reduction in the overall speedup across all target models, however, due to its enhanced potential capability, the acceptance rate does improve across all the target models.

### 4.3.2 N-gram Cache Memory Footprint

We also analyzed the various cache size at the end of training for each of the reported tasks as shown in Table 4. Overall n-gram cache size across tasks remain relatively small compared to the size of the draft or target model. This indicates potential feasibility for on-device setting given the small overhead.

### 4.3.3 Effectiveness of N-gram Cache

In this section, we focus on the impact of the N-gram cache to different training variants as per Table 5. We perform ablation on a subset of GSM8K dataset (4k samples) across multiple techniques. It can be seen that the baseline $\text{SpD}_{DM}$ performs poorly which indicates the mismatch between the drafter and the target model alignment for the pretrained models. We also capture all the possible N-gram matches during the $\text{SpD}_{DM}$ baseline for the train set, and then use the cache as a lookup for an improved baseline in the $\text{SpD}_{DM} + \text{N-gram}_{pp}$. As such there is a cache hit of 0.87, which indicates that the baseline model without any pretraining can still benefit with the N-gram cache. We also train the model to improve the alignment using $\mathcal{L}_{DM}$ and although without the N-gram, we can still see a large improvement over the baseline. Moreover. when we train with N-gram$_{pp}$, there is a further improvement on the overall speedup. Finally, we train using both $\mathcal{L}_{DM} + \lambda\mathcal{L}_{N\text{-gram}}$, which provides the best results across all techniques on different draft length $k$.

Table 3: Comparison of Cross-vocabulary Distillation Performance with Llama-68M and Llama-160M across three target models

| Target | Method | GSM8K | | MBPP+HumanEval | |
|---|---|---|---|---|---|
| | | Acc Rate | Speedup | Acc Rate | Speedup |
| Qwen2.5 7B | $\text{SpD}_{DM}$ (68M) | 0.15 | 1.02x | 0.10 | 0.94x |
| | $\mathcal{L}_{\text{DM}} + \lambda\mathcal{L}_{\text{N-gram}}$ (68M) | **0.401** | **1.66x** | **0.27** | **1.33x** |
| | $\text{SpD}_{DM}$ (160M) | 0.18 | 0.70x | 0.13 | 0.60x |
| | $\mathcal{L}_{\text{DM}} + \lambda\mathcal{L}_{\text{N-gram}}$ (160M) | **0.47** | **1.12x** | **0.32** | **0.90x** |
| Qwen2.5 14B | $\text{SpD}_{DM}$ (68M) | 0.15 | 1.17x | 0.10 | 1.14x |
| | $\mathcal{L}_{\text{DM}} + \lambda\mathcal{L}_{\text{N-gram}}$ (68M) | **0.407** | **1.92x** | **0.272** | **1.57x** |
| | $\text{SpD}_{DM}$ (160M) | 0.178 | 0.89x | 0.13 | 0.84x |
| | $\mathcal{L}_{\text{DM}} + \lambda\mathcal{L}_{\text{N-gram}}$ (160M) | **0.472** | **1.40x** | **0.33** | **1.19x** |
| Qwen2.5 32B | $\text{SpD}_{DM}$ (68M) | 0.153 | 1.30x | 0.10 | 1.23x |
| | $\mathcal{L}_{\text{DM}} + \lambda\mathcal{L}_{\text{N-gram}}$ (68M) | **0.42** | **2.05x** | **0.274** | **1.71x** |
| | $\text{SpD}_{DM}$ (160M) | 0.187 | 1.03x | 0.133 | 0.97x |
| | $\mathcal{L}_{\text{DM}} + \lambda\mathcal{L}_{\text{N-gram}}$ (160M) | **0.49** | **1.62x** | **0.335** | **1.40x** |

Table 4: Summary of final N-gram cache size across tasks with Llama-68M drafter and Qwen2-7B target after online inference and distillation. Cache memory (MB) is derived from pympler.asizeof.asizeof().

| | GSM8K | MBPP+HumanEval | Alpaca | XSUM |
|---|---|---|---|---|
| Training Samples | 7473 | 910 | 8000 | 4000 |
| Cache Size (#n-grams) | 5569 | 2238 | 20339 | 17013 |
| Cache Memory (MB) | 1.372 | 0.501 | 4.569 | 3.924 |

### 4.3.4 Distillation Loss Comparisons

The different loss variants also provide different levels of performance on the test set as shown in Table 6. When trained only on the $\mathcal{L}_{\text{N-gram}}$, the training is very unstable. This could either be due to the number of n-grams being substantially smaller than the direct mapping tokens within most datasets, or the training requires additional constraints to direct towards a minima. Consequently, training on the combined loss provides better performance metrics across temperatures as well. The final $\mathcal{L}_{\text{DM}}$ KL + $\lambda\mathcal{L}_{\text{N-gram}}$ KL, 2, provides slightly better results on temperature = 1 indicating the impact of the additional KL over the intersection vocabulary which is beneficial for the sampling process. However, we noticed tuning the scaling factor $\lambda$ becomes critical and hence we use Equation 1 for all of the experiments since it was much more stable across all tasks. We fix a $\lambda = 0.2$ for all the tasks, across all experiments.

### 4.3.5 Adaptive Drafter Initialization and Thresholds

Two additional aspects in training and using adaptive drafting are the acceptance prediction head initialization and the stopping threshold for early exit. Before the joint training of model distillation and adaptive drafting, we can pretrain the acceptance prediction head on an offline dataset, while keeping the drafter fixed. This should ideally capture some priors of the draft-target alignment and provide a better initialization for the online joint training. To verify this, we pretrain the acceptance prediction head on the Alpaca dataset for 1 and 3 epochs respectively, and use the two checkpoints as initialization for online training. We observe mixed results where the pretrained initializations harm performance on GSM8K and XSum, and only improve on MBPP+HumanEval by small margins. We suspect MBPP+HumanEval has a closer affinity in data distribution to Alpaca while the other two do not, which leads to negative transfer with pretraining.

The stopping threshold $\gamma$ in adaptive drafting also plays a key role for speedup performance. In Alpaca and XSum we observe a conservative threshold of $\gamma = 0.3$ is sufficient to attain better speedup than baselines, while in GSM8K and MBPP+HumanEval we need a more relaxed threshold

Table 5: Effect of n-grams cache with Llama-68M and Llama3-8B on GSM8K (subset)

| Metrics | $\text{SpD}_{DM}$ | $\text{SpD}_{DM}$ + N-gram$_{pp}$ | $\mathcal{L}_{\text{DM}}$ | $\mathcal{L}_{\text{DM}}$ + N-gram$_{pp}$ | $\mathcal{L}_{\text{DM}}$ + $\lambda\mathcal{L}_{\text{N-gram}}$ |
|---|---|---|---|---|---|
| | | | $k = 3$ | | |
| Acc Rate | 0.16 | 0.20 | 0.40 | 0.42 | **0.46** |
| Speedup | 1.04x | 1.16x | 1.59x | 1.61x | **1.66x** |
| Avg n-gram hit | 0 | 0.87 | 0 | 0.48 | **2.40** |
| | | | $k = 4$ | | |
| Acc Rate | 0.12 | 0.16 | 0.32 | 0.35 | **0.41** |
| Speedup | 1.01x | 1.11x | 1.51x | 1.54x | **1.61x** |
| Avg n-gram hit | 0 | 1.49 | 0 | 0.75 | **3.66** |

Table 6: Training Loss comparisons with Llama-68M and Llama3-8B on GSM8K (subset)

| Metrics | $\mathcal{L}_{\text{N-gram}}$ NLL | $\mathcal{L}_{\text{DM}}$ NLL + $\mathcal{L}_{\text{N-gram}}$ NLL | $\mathcal{L}_{\text{DM}}$ KL + $\mathcal{L}_{\text{N-gram}}$ NLL | $\mathcal{L}_{\text{DM}}$ KL + $\mathcal{L}_{\text{N-gram}}$ KL |
|---|---|---|---|---|
| | | Temperature $= 0.01$ | | |
| Acc Rate | 0.090 | 0.368 | 0.375 | **0.376** |
| Speedup | 0.86x | 1.51x | **1.57x** | **1.57x** |
| | | Temperature $= 1$ | | |
| Acc Rate | 0.070 | 0.271 | 0.273 | **0.275** |
| Speedup | 0.79x | 1.23x | 1.27x | **1.31x** |

of $\gamma = 0.7$ to achieve comparable speedup. We also observe applying a relaxed threshold on model trained on a stricter threshold improves speedup performance, indicating adaptive drafting could be prone to underestimating the maximum acceptance draft length. It also suggests the actual stopping threshold to be used for inference should be chosen based on the task and then adjusted based on online performance.

## 4.4 Limitations

While most of our results indicate the potential of our methodology and the OmniDraft framework, some potential limitations still require additional research. 1) Although we are training on incoming data for online adaptation of the drafter, since it is limited to a single iteration of the data stream, there is still potential for instability on new unseen data. 2) We currently use the full n-gram cache per task per target model, however, should memory become a bottleneck, it would require optimized cache eviction policy to cater to edge devices. 3) Special tokens that do not have direct mapping currently would require some additional effort to handle. Consequently, this would make it less seamless to integrate multi-modal tasks. As part of our future plan of action, although cross-vocabulary speculative decoding already implicitly includes the adaptive proposal length due to n-gram merge, we are working towards incorporating an explicit adaptive head in the cross-vocabulary setting.

## 5 Conclusion

In this work we propose the OmniDraft framework that leverages n-gram cache and hybrid distillation loss to enable cross-vocabulary speculative decoding. We show how the draft model can be aligned to different target models with different vocabulary space via online adaptation, and we further showcase online adaptive drafting to get additional speedup. Our empirical results also show good performance across all metrics. Overall, OmniDraft shows great potential and could pave way to new on-device LLM applications.

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

# A  Implementation Details

This appendix provides detailed information about the implementation and training setup used in our experiments.

## A.1  Model and Hardware Details

Throughout our work, we use the environment setup with NVIDIA A100 GPU (40/80GB), PyTorch 2.1.0 framework, CUDA version 12.1, and Ubuntu 22.04 LTS. Our models used include Llama-68M [32], Llama3-8B [17], Qwen2-7B [49] and Vicuna-7B [52]. We train the Llama-68M model in our experiments. Their model details are listed in the following.

Table 7: Model hyperparameters

| Hyperparameter | Value |
|---|---|
| model | Llama-68M |
| layers | 2 |
| hidden size | 768 |
| attention heads | 12 |
| activation function | SiLU |

Table 8: Model latency and vocabulary statistics

| Model | Wall-time per Step (s) | Vocabulary Size | Intersection with Llama-68M |
|---|---|---|---|
| Llama-68M | 0.00203 | 32000 | 32000 |
| Qwen2-7B | 0.02667 | 152064 | 22275 |
| Llama3-8B | 0.02846 | 128256 | 22416 |
| Vicuna-7B | 0.02683 | 32000 | 32000 |

## A.2  Training Details

Our experiments use GSM8K [12], Alpaca [44], XSum [36] and a combined MBPP+HumanEval [4][10] datasets. We also show the major hyperparameters used across the experiments. For evaluation, we perform three runs with different seeds and report the average in the main results sections.

Table 9: Dataset details

| Dataset | GSM8K | MBPP+HumanEval | Alpaca | XSum |
|---|---|---|---|---|
| train | 8K | 1K | 8K | 4K/8K |
| test | 200 | 228 | 100 | 100 |

Table 10: Training hyperparameters

| Hyperparameter | Value |
|---|---|
| batch size | 8 |
| learning rate (LR) | 1e-4/2e-5 |
| LR scheduler | constant |
| optimizer | AdamW |
| $\beta_1$ | 0.9 |
| $\beta_2$ | 0.999 |
| weight decay | 0.01/0 |
| epochs | 1 (online) |
| mixed precision | FP16 |
| LoRA rank | 32 |
| temperature | 0.01 |

# B  Additional Experiment Details

## B.1  Speedup Metric

We follow the convention in Medusa [8] to define the speedup metric. Given **acceleration rate** as the average number of tokens decoded per decoding step and **overhead** as the average per step latency of the proposed model divided by that of the vanilla model, **speedup** refers to the wall-time acceleration rate and can be computed with **speedup** = **acceleration rate**/**overhead**.

## B.2  Adaptive Drafting Training

We show the training curves for adaptive drafting across all tasks. It can be seen that our proposed variants of joint or interleaved align + adapt training have the best performances in both average acceptance rate ans speedup.

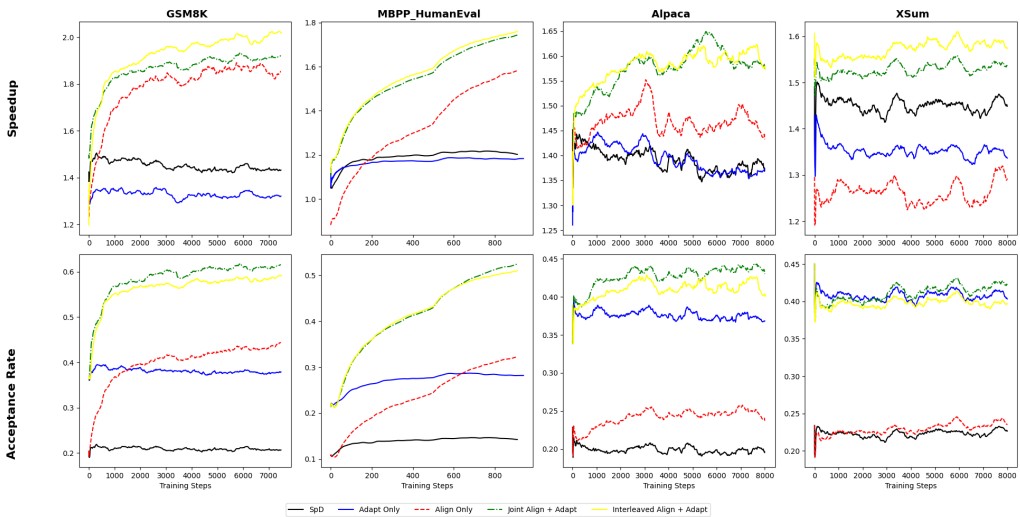

Figure 3: Online Adaptive Drafting training plot of Llama-68M vs Vicuna-7B

## B.3  Adaptive Drafting LoRA Training

To showcase the idea of *"one drafter for all"*, we further show results of online adaptive drafting with LoRA used in distillation in Table 11. We observe slightly lower performances across tasks compared to distillation with full fine-tuning as expected. But each of the distillation or distillation with adaptive drafting baseline with LoRA still outperforms normal speculative decoding. And distillation with adaptive drafting outperforms the distillation only baseline, except on GSM8K where the gap is minimal.

By demonstrating the compatibility of LoRA with our proposed online distillation and adaptive drafting training, we empower a single draft model to serve as the backbone and selectively fine-tune LoRA modules and acceptance prediction heads for any potential target model. This would strike a good balance to keep minimal memory overhead while retaining the benefits of greater speedup and flexibility, which are ideal for on-device applications.

Table 11: Performance on Online Adaptive Drafting using LoRA with rank 32 with Llama-68M and Vicuna-7B.

| Target | Method | GSM8K | | MBPP+HumanEval | | Alpaca | | XSum | |
|---|---|---|---|---|---|---|---|---|---|
| | | Acc Rate | Speedup | Acc Rate | Speedup | Acc Rate | Speedup | Acc Rate | Speedup |
| | SpD | 0.21 | 1.44x | 0.14 | 1.22x | 0.20 | 1.44x | 0.20 | 1.42x |
| Vicuna-7B | LoRA Distill Only | 0.37 | **1.95x** | 0.24 | 1.52x | 0.25 | 1.54x | 0.23 | 1.49x |
| | Joint LoRA Distill + Adapt | **0.49** | 1.87x | **0.39** | 1.59x | 0.39 | **1.58x** | **0.41** | 1.54x |
| | Interleaved LoRA Distill + Adapt | **0.49** | 1.94x | 0.38 | **1.61x** | **0.42** | **1.58x** | 0.39 | **1.55x** |

### B.4 Cross-vocabulary SpD online distillation on Llama-68M with Llama3-8B as target

Figure 4 showcases the training dynamics of Llama-68M with Llama3-8B as the target model. Similar to the previous Figure 2, the pattern is very much matched. Overall, across all the tasks, our methods are able to improve upon the SpD baseline significantly as training progresses.

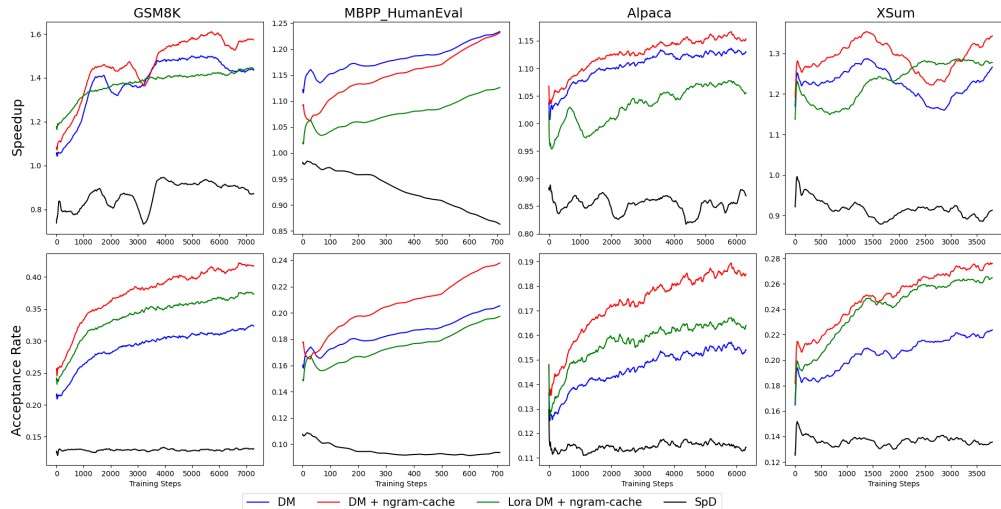

Figure 4: Cross-vocabulary SpD online distillation on Llama-68M with Llama3-8B as target

### B.5 Rank ablation for Cross-vocabulary online distillation of LoRA

We also ablate on the rank for the LoRA experiments, specifically focussed on the $\mathcal{L}_{DM} + \lambda\mathcal{L}_{N\text{-gram}}$ + LoRA method. As shown in the Table 12, it is clear that as the rank increases, performance also improves, but beyond rank 32, there is a diminishing return in terms of overall improvement. As such we choose a conservative rank of 32 across all our experiments, tasks and methodologies. Moreover, for the *"one drafter for all"* setting, folding of the LoRA weights is infeasible and as such having a lower rank would benefit the overall memory and compute required for the drafter model on-device.

Table 12: Effect of rank for $\mathcal{L}_{DM} + \lambda\mathcal{L}_{N\text{-gram}}$ + LoRA for Llama-68M and Llama3-8B on GSM8K

| Metrics \ LoRA Rank | 8 | 16 | 32 | 64 | 128 |
|---|---|---|---|---|---|
| Acc Rate | 0.30 | 0.34 | 0.37 | 0.38 | **0.38** |
| Speedup | 1.478x | 1.543x | 1.595x | 1.625x | **1.632x** |

### B.6 Distribution Shift

#### B.6.1 Heterogeneous Dataset Shift

We also ablated on dataset distribution shift and data heterogeneity. We analyzed our training framework by starting the training on the GSM8K dataset and after around 1k steps, shifting to a new data distribution of MBPP+HumanEval Mix. As shown in Figure 5, initially, both speedup and acceptance rate show a steady upward trend, indicating that the model is learning effectively and becoming more efficient. At step 1000, we introduce a new dataset with a different distribution which causes a sharp drop in both acceptance rate and speedup, highlighting the impact of this shift. However, following this disruption, both metrics begin to gradually recover, demonstrating the model's ability to adapt to the new data distribution over time. Additionally, when using our n-gram cache, the recovery seems to be much better and larger when compared to no cache. Finally, when comparing to the model that was solely trained on MBPP+HumanEval mix dataset, the model is able to slowly reach similar performance on both metrics. Overall, using our n-gram cache and despite the distribution shift, the model is showing resilience and adaptability to the new dataset.

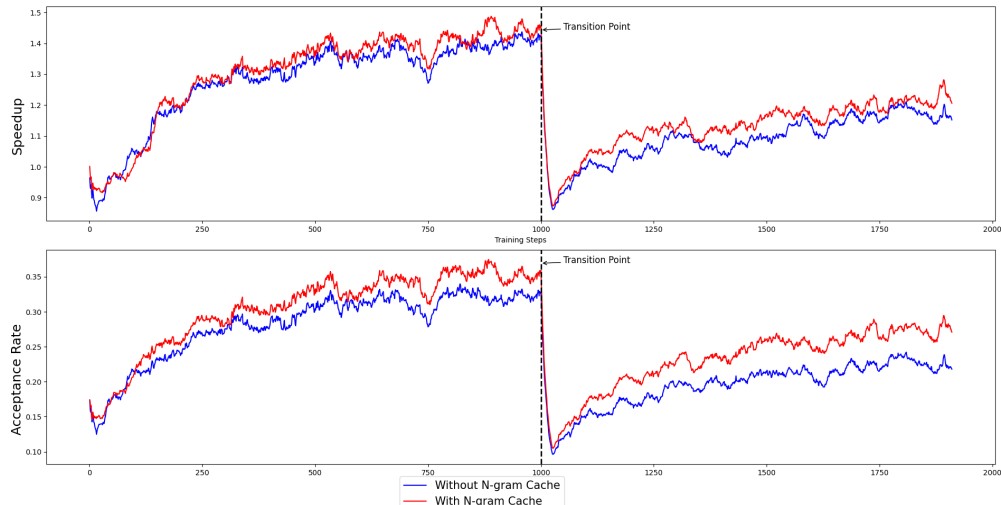

Figure 5: Dataset Drift tracking during training by switching the dataset from GSM8K to MBPP+HumanEval at Step 1000 for Llama-60M vs Qwen-2.5-7B model

### B.6.2   Target Model Shift

To further understand the impact of target model drift during training, We conducted empirical analysis on target switching between Qwen2.5 7B and 14B at arbitrary step intervals (e.g., 14B → 7B → 14B) As shown in Figure 6, our observations indicate that the drafter model is able to re-align seamlessly after each switch. Notably, switching to the 7B model yields reduced speedup due to its lower drafter latency, while maintaining alignment quality. This behavior is largely attributed to the fact that the target models belong to the same model family and as such the drafter is able to continually align during training. We also evaluated the impact of enabling the n-gram cache during target switching. Results show that the drafter recovers alignment quickly when the cache is enabled. However, in scenarios involving cross-family model swaps, a new n-gram cache is preferred due to differences in tokenization. In such cases, a simple distribution shift is insufficient to maintain alignment.

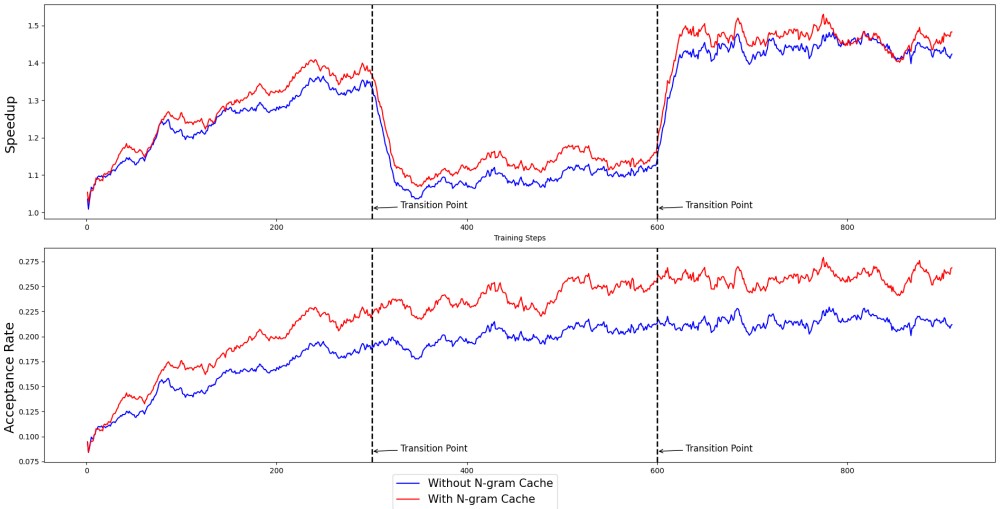

Figure 6: Model Drift tracking during training by switching the target model from 14B to 7B and back to 14B at step 300 and step 600, with Llama-68M as drafter

# C Additional Algorithm Details

We show the detailed algorithm for cross-vocabulary distillation in Algorithm 2. The online distillation training leverages our proposed cross-vocabulary speculative decoding in inference or data collection, and uses the hybrid distillation loss to update the draft model.

---
**Algorithm 2** Cross-vocabulary Distillation

---
1: Given target $p(\cdot)$, drafter $q_\theta(\cdot)$, online data stream $\mathcal{S}$, data buffer $\mathcal{Q}$, update interval $I$.
2: $i, \mathcal{Q} \leftarrow 0, \{\}$
3: **while** True **do**
4:     $x \sim \mathcal{S}, i \leftarrow i + 1$
5:     Get a response $y$ with cross-vocabulary speculative decoding as in Algorithm 1
6:     Append $(x, y)$ to $\mathcal{Q}$
7:     **if** $i \mod I == 0$ **then**
8:         Update $q_\theta$ on $\mathcal{Q}$ with hybrid distillation loss $\mathcal{L}_{\text{cross\_vocab\_distill}}(\theta)$
9:         $\mathcal{Q} \leftarrow \{\}$
10:    **end if**
11: **end while**

---

We show the detailed algorithms for the two variants of online adaptive drafting training in Algorithm 3 and 4. The distillation loss $\mathcal{L}_{\text{distill}}(\theta)$ uses KL divergence on generated response tokens. The adaptive drafting loss $\mathcal{L}_{\text{adapt}}(\phi)$ is a weighted BCE loss between the acceptance prediction head outputs and acceptance ratio labels constructed from the target model $p$ and current draft model $q_\theta$.

---
**Algorithm 3** Online Adaptive Drafting — Joint Training

---
1: Given target $p(\cdot)$, drafter $q_\theta(\cdot)$, acceptance prediction head $f_\phi(\cdot)$, online data stream $\mathcal{S}$, data buffer $\mathcal{Q}$, update interval $I$.
2: $i, \mathcal{Q} \leftarrow 0, \{\}$
3: **while** True **do**
4:     $x \sim \mathcal{S}, i \leftarrow i + 1$
5:     Get a response $y$ with speculative decoding using adaptive drafting
6:     Push $(x, y)$ to $\mathcal{Q}$
7:     **if** $i \mod I == 0$ **then**
8:         Update $q_\theta$ on $\mathcal{Q}$ with distillation loss $\mathcal{L}_{\text{distill}}(\theta)$
9:         Compute labels $l = \min(1, p(y)/q_\theta(y))$ on $(x, y) \in \mathcal{Q}$
10:        Update $f_\phi$ on $\{(x, y, l)\}_{|\mathcal{Q}|}$ with adaptive drafting loss $\mathcal{L}_{\text{adapt}}(\phi)$
11:       $\mathcal{Q} \leftarrow \{\}$
12:    **end if**
13: **end while**

---

# D Cross-vocabulary N-gram Cache Ablations

## D.1 N-gram distributions

To showcase the captured n-grams during our cross-vocabulary speculative decoding, we collect the n-gram cache across tasks after online training with the hybrid distillation losses. We plot the distribution of the n-gram counts with respect to the frequencies they are encountered in Figure 7. We include n-grams with maximum frequency of 3000 and use a log-scale of the n-gram counts for cleaner visualization. N-grams with higher frequencies over 3000 typically represents a group of delimiters such as spaces and newline characters that are used for formatting. Besides these, we can observe a clear long-tail distribution for more frequent n-grams. Although the majority of n-grams have low frequencies which are not guaranteed to re-appear in future data stream, the long-tail frequency n-grams still have non-trivial contribution and can be utilized to speed up the cross-vocabulary speculative decoding process. We also notice the frequent n-grams constitute a higher percentage in the MBPP+HumanEval domain, despite having smaller dataset size. It indicates

**Algorithm 4** Online Adaptive Drafting — Interleaved Training

---

1: Given target $p(\cdot)$, drafter $q_\theta(\cdot)$, acceptance prediction head $f_\phi(\cdot)$, online data stream $\mathcal{S}$, distillation data buffer $\mathcal{Q}$, adaptive drafting data buffer $\mathcal{R}$ with max size $N$, batch size $B$, update interval $I$.
2: $i, \mathcal{Q}, \mathcal{R} \leftarrow 0, \{\}, \{\}$
3: **while** True **do**
4:    $x \sim \mathcal{S}, i \leftarrow i + 1$
5:    Get a response $y$ with speculative decoding using adaptive drafting
6:    Push $(x, y)$ to $\mathcal{Q}$
7:    **if** $i \mod I == 0$ **then**
8:       Update $q_\theta$ on $\mathcal{Q}$ with distillation loss $\mathcal{L}_{\text{distill}}(\theta)$
9:       Push $\mathcal{Q}$ to $\mathcal{R}$
10:      **if** $|\mathcal{R}| > N$ **then**
11:         Evict early data from $\mathcal{R}$
12:      **end if**
13:      $\mathcal{Q} \leftarrow \{\}$
14:    **else**
15:      Sample a batch $\mathcal{R}_B \in \mathcal{R}$
16:      Compute labels $l = \min(1, p(y)/q_\theta(y))$ on $(x, y) \in \mathcal{R}_B$
17:      Update $f_\phi$ on $\{(x, y, l)\}_B$ with adaptive drafting loss $\mathcal{L}_{\text{adapt}}(\phi)$
18:    **end if**
19: **end while**

---

the n-gram cache technique for cross-vocabulary speculative decoding can be more effective in well-structured domain like coding.

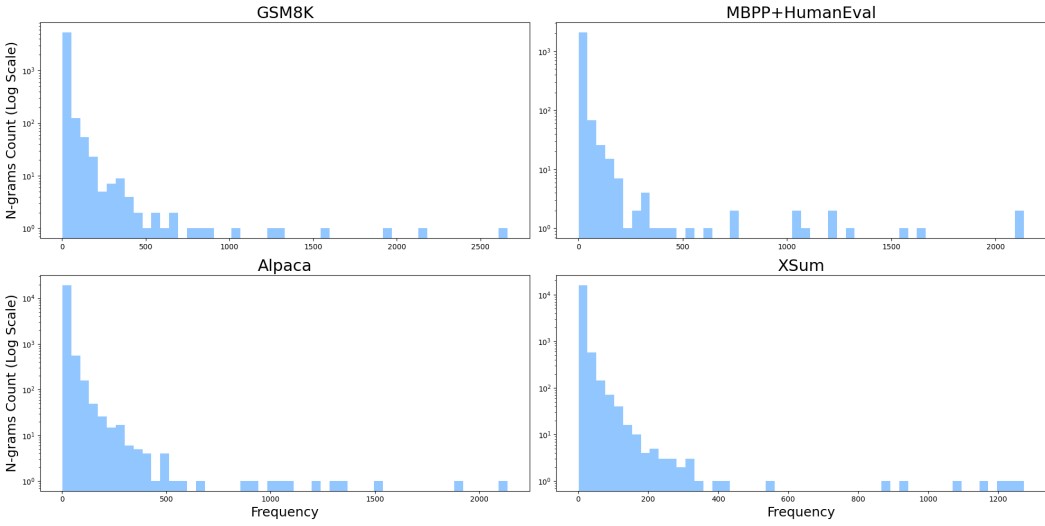

Figure 7: Cross-vocabulary N-gram Distributions of Llama-68M and Qwen2-7B

## D.2 N-gram inference examples

Next in Table 13, we show examples of the n-grams being used from the cache during inference in different tasks. The n-gram examples are extracted from actual samples in the datasets with Llama-68M and Qwen2-7B, given a draft length of 4 each proposal or speculative decoding step. Pink tokens represent tokens with direct mapping between the drafter vocabulary and target vocabulary. Yellow tokens represent sub-tokens in the drafter space and the corresponding n-gram tokens in the target space, which are mapped via cache lookup. Lime tokens represent those that are accepted by the target model after verification. Note that even with the mapping, the proposed n-gram tokens are not guaranteed to be accepted and require cross-vocabulary distillation to further align their distributions with the target.

| Proposed Tokens (Drafter Vocab) | Mapped Tokens (Target Vocab) | Accepted Tokens |
|---|---|---|

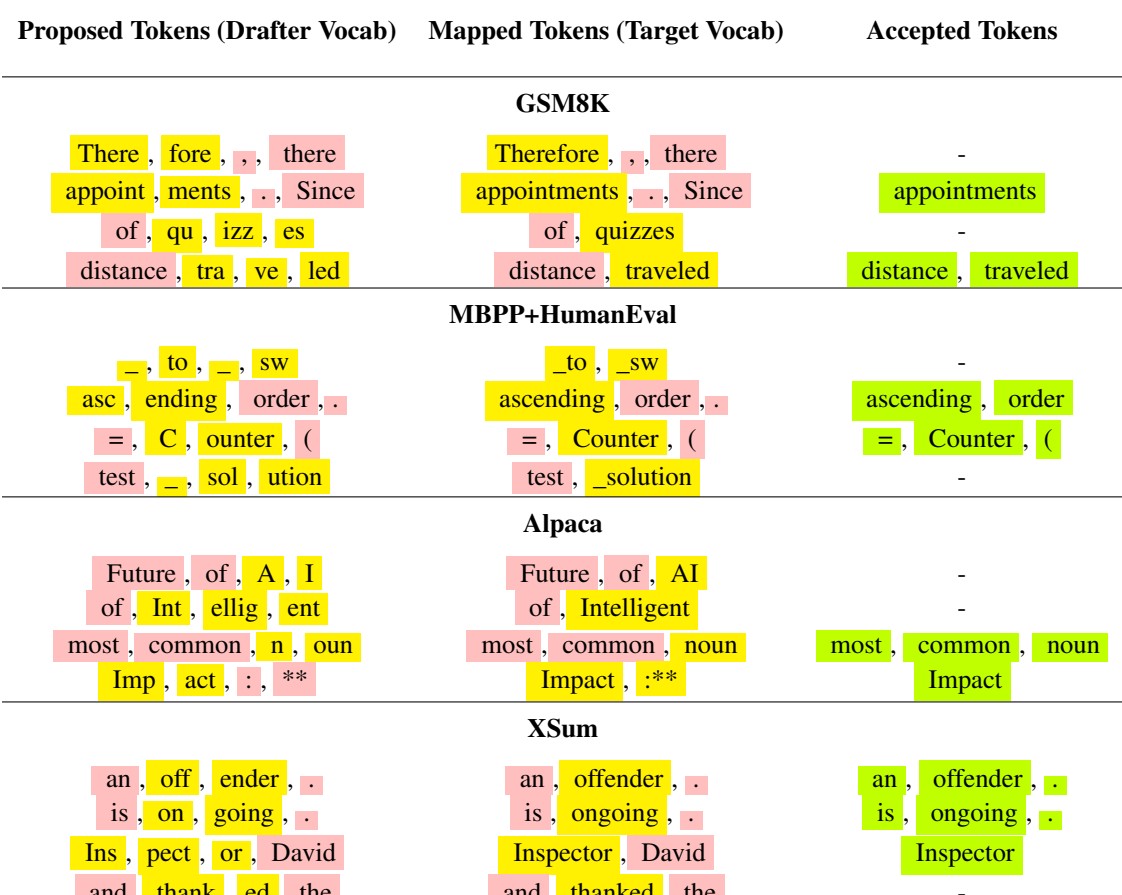

**GSM8K**

| There , fore , , , there | Therefore , , , there | - |
| appoint , ments , . , Since | appointments , . , Since | appointments |
| of , qu , izz , es | of , quizzes | - |
| distance , tra , ve , led | distance , traveled | distance , traveled |

**MBPP+HumanEval**

| _ , to , _ , sw | _to , _sw | - |
| asc , ending , order , . | ascending , order , . | ascending , order |
| = , C , ounter , ( | = , Counter , ( | = , Counter , ( |
| test , _ , sol , ution | test , _solution | - |

**Alpaca**

| Future , of , A , I | Future , of , AI | - |
| of , Int , ellig , ent | of , Intelligent | - |
| most , common , n , oun | most , common , noun | most , common , noun |
| Imp , act , : , ** | Impact , :** | Impact |

**XSum**

| an , off , ender , . | an , offender , . | an , offender , . |
| is , on , going , . | is , ongoing , . | is , ongoing , . |
| Ins , pect , or , David | Inspector , David | Inspector |
| and , thank , ed , the | and , thanked , the | - |

Table 13: Examples of N-gram Cache Across Tasks. Each row represents one cross-vocabulary speculative decoding step/proposal extracted from the inference results on the data test set (no sequential order between rows). Drafter proposes 4 tokens each time, which are mapped to the target space either as direct mapping tokens or as merged n-gram tokens . Target model then verifies the mapped tokens to get the final accepted tokens , plus any correction token from the residual distribution or a sampled free token if all mapped tokens are accepted. We also denote no token being accepted as "-".

## D.3 N-gram learning

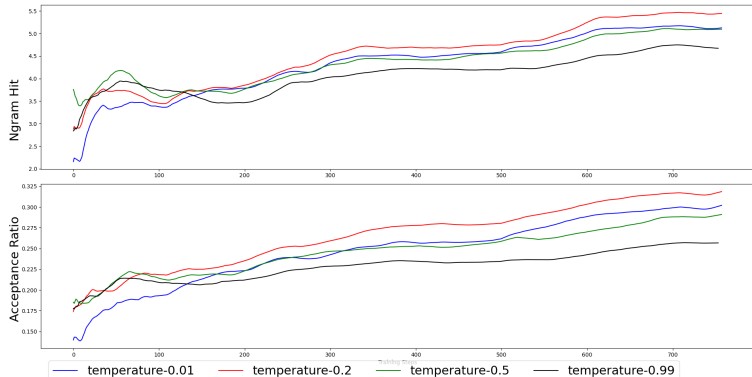

Figure 8: Evolution of n-gram tokens learning with Llama-68M and Llama3-8B on **MBPP+HumanEval** dataset. **N-gram hit** refers to the number of n-grams proposed by drafter in each response, averaged over query samples. **Acceptance ratio** refers to the quantity $\min(1, p/q)$ in speculative decoding, it is averaged over proposal steps with an n-gram token and is indicative of the distribution alignment between the drafter and target. These two metrics show cross-vocabulary distillation learns to slowly align n-gram distributions to the target and utilize them more in inference.

From Figure 8, it can be seen that during the training of $\mathcal{L}_{\text{DM}} + \lambda \mathcal{L}_{\text{N-gram}}$ on **MBPP+HumanEval** dataset, the average acceptance ratio for the n-grams is steadily improving, showing the impact of the loss function for better alignment of the n-grams. Furthermore, the impact is clearly visible even across different temperatures. Similarly as training progresses, the average number of successful cache hit for n-gram tokens per query is also improving.

## D.4 N-gram Cache Growth Rate

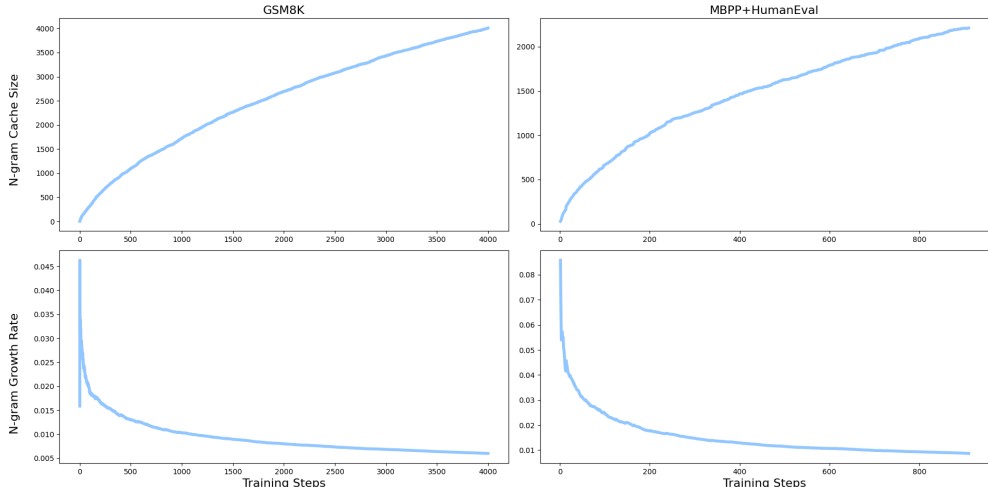

Figure 9: N-gram cache growth rate for Llama-68M drafter and Qwen2-7B target on GSM8K (4K subset) and MBPP+HumanEval datasets.

From Figure 9, We can define the growth rate of the N-gram cache to be growth_rate = cache_size/training_tokens. From the tables, we can observe that the cache grows sub-linearly with number of tokens processed in online learning. The growth rate of the N-gram cache can also be seen to continually decrease.

### D.5 N-gram Cache Scalability

Table 14 indicates how speedup is impacted by different max cache size and cache eviction policies: Least Recently Used (LRU) that evicts the n-gram that hasn't been accessed for the longest time, Least Frequently Used (LFU) that removes the n-gram with the lowest access frequency. "1/4" and "1/2" refer to using max cache size as a quarter and half of the full final cache size that we previously trained.

Among all eviction policies and max cache sizes, both evaluation metrics are better than using no cache. Specifically LRU shows diminishing returns over the growth of the cache. Meanwhile LFU plateaus at an earlier stage, with performance matching the full cache even with a quarter of the size. This could be due to LFU capturing most of the high frequency and important n-grams, compared to LRU which only captures the recent n-grams. In the context of on-device deployment, this implies with proper selection of cache size and eviction strategy, we can potentially attain robust performance given only constrained memory resources, highlighting the practical feasibility of our approach.

Table 14: Cache size scaling on MBPP+HumanEval

| Cache Size | Cache Type | Acc Rate | Speedup |
|---|---|---|---|
| No Cache | - | 0.219 | 1.29x |
| 1/4 | LRU | 0.254 | 1.33x |
|  | LFU | 0.259 | 1.36x |
| 1/2 | LRU | 0.261 | 1.35x |
|  | LFU | 0.263 | 1.36x |
| Full Cache | - | 0.267 | 1.36x |

