# OpenReview forum: "OmniDraft: A cross-vocabulary, online adaptive drafter for on-device speculative decoding"
_NeurIPS.cc/2025/Conference — NeurIPS 2025 poster_

### Official Review · Reviewer_PzLn · 2025-06-29

**Clarity:** 3
**Significance:** 4
**Originality:** 2
**Rating:** 4
**Confidence:** 3

**Summary:**

This paper proposes a new framework for speculative decoding (termed as OmniDraft) that addresses two major challenges in online deployment settings - cross-vocabulary incompatibility between draft and target models, and the need for dynamic latency improvements over usage time. OmniDraft proposes an online n-gram cache with hybrid distillation fine-tuning to handle vocabulary mismatches and leverages adaptive drafting techniques to further improve decoding speed.

The overall motivation of this work is to obtain a "one ring rules all" type framework - enable a single, small draft model to work effectively with various larger target models for speculative decoding in on-device LLM applications. The authors show empirically that their generalized idea indeed works in practice and gives 1.5-2x speedup on tasks like math reasoning, coding, and text generation.

**Questions:**

- Beyond the mentioned future work of incorporating an explicit adaptive head in the cross-vocabulary setting, what other avenues do the authors have in mind to handle special tokens?

- Could you elaborate on the potential strategies for optimizing the n-gram cache for memory-constrained edge devices? (e.g. offload, caching, indexing, etc)

- Do the authors see any other drawbacks with the "dynamic adapter switching" approach e.g. for LoRA work in practice?

**Ethical Concerns:**

["NO or VERY MINOR ethics concerns only"]

**Limitations:**

N/A (no data or models released).

**Paper Formatting Concerns:**

Minor typos and language issues found. Encourage authors to run grammar check on their draft. Highlighting a few below:

Line 210: "or we slice out the a portion of the train set as the test set."
"the a portion" should be "a portion".

Line 250: "This could be due the difficulty in training the acceptance prediction head in an online setting."
"due the" should be "due to the".

Line 258: "acceptance probability, leading to wrongful early exit."
"wrongful early exit" could be rephrased as "incorrect early exit" or "premature early exit." to make it sound better.

**Quality:**

2

**Strengths And Weaknesses:**

Strengths:
- The paper tackles two significant practical challenges in speculative decoding: cross-vocabulary mismatch and online adaptation for latency improvement, both of which are crucial for real-world on-device LLM deployment. In this regard, this is a good contribution.
- There are some technical strong ideas in this paper such as the introduction of an online n-gram cache for cross-vocabulary translation which I find a very effective solution to the vocabulary mismatch problem - perhaps there are more efficient ones but this is a reasonable practical approach imo.
- Integration of online adaptive drafting is another further improvement in the proposed method - this makes their approach more flexible by allowing the drafter to dynamically adjust proposal length.

Weaknesses:
First of all, I applaud the authors for thinking through and bringing out several possible limitations of their work. Many of my thoughts are in the same direction, although I will re-emphasize:
- The paper acknowledges that online adaptation, being limited to a single iteration of the data stream, still has potential for instability on new unseen data. This is an inherent challenge of online learning but remains a practical limitation.
- Is the draft model indeed universal? How does it work in the presence of distribution shifts and data heterogeneity? I feel the discussion on this is limited in this paper but compelling direction to make such methods work in practice.

---

> ### Author Rebuttal · Authors · 2025-07-31
>
> We would like to express our gratitude to the reviewers for their careful review of our paper and thoughtful feedback. We appreciate the opportunity to clarify the contributions, novelty and limitations of our work. We are thankful to reviewer PzLn for recognizing the novelty and practicality of our approach.
>
>
> ### Q1: Instability from Online Adaptation, Performance on Distribution Shifts and Data Heterogeneity
>
> #### Online Learning
> - **Clarifications**: Regarding online learning, the observed instability is an inherent characteristic of the setting, which occurs as fluctuations (e.g., zig-zag patterns) in the training curves. However, as training progresses and the n-gram cache becomes more populated and captures a broader set of in-distribution data, the instability can be potentially mitigated by training with more samples. We can also incorporate other techniques such as replay buffer or other aggressive regularization techniques for online learning instability mitigation.
>
> #### Distribution Shifts
> - **Clarifications**: Regarding the dataset distribution shift and data heterogeneity, we analyzed our training framework by starting the training on the GSM8K dataset and after around 1k steps, shifting to a new data distribution of MBPP+HumanEval Mix.
> - **Reasoning**: As shown in `Table 1`, initially, both speedup and acceptance rate show a steady upward trend, indicating that the model is learning effectively and becoming more efficient. At step 1000, we introduce a new dataset with a different distribution which causes a sharp drop in both acceptance rate and speedup, highlighting the impact of this shift. However, following this disruption, both metrics begin to gradually recover, demonstrating the model's ability to adapt to the new data distribution over time. Additionally, when using our n-gram cache, the recovery seems to be much better and larger when compared to no cache. Finally, when comparing to the model that was solely trained on MBPP+HumanEval mix dataset, the model is able to slowly reach similar performance on both metrics. Overall, using our n-gram cache and despite the distribution shift, the model is showing resilience and adaptability to the new dataset. We will be updating the final paper with these additional plots in the appendix.
>
> Table 1. Performance with Data Distribution Shift
>
> |**Phase**|**Step Range**|**Model**|**Acceptance Rate**|**Speedup**|
> |-|-|-|-|-|
> |Initial Training (GSM8K)|0 – 1000|With Cache|Increasing ~0.15 - ~0.35|Increasing ~1.0x - ~1.45x|
> |||No Cache|Increasing ~0.14 - ~0.32|Increasing ~0.96x - ~1.42x|
> |Dataset Switch (MBPP+HumanEval)|1000|With Cache|Sharp Drop ~0.11|Sharp Drop ~0.87x|
> |||No Cache|Sharp Drop ~0.09|Sharp Drop ~0.86x|
> |Recovery (MBPP+HumanEval)|1000 – End|With Cache|Gradual Increase to ~0.28|Gradual Increase to ~1.22x|
> |||No Cache|Gradual Increase to ~0.22|Gradual Increase to ~1.16x|
>
>
> ### Q2: Special Tokens Handling
>
> - **Clarifications**: Thank you for the question. Regarding special tokens across models with different vocabularies, we treat them as regular text string during the processing in the drafter. For example, tokens like `<im_start>` and `<im_end>` are handled using the drafter's tokenizer and encoded as part of the n-gram sequence. As training progresses, the drafter learns to associate these token patterns with their intended semantics, effectively learning the appropriate mappings.
> - As part of future work of extending our technique to multimodal models which might have a larger number of special tokens, if the draft model has unused reserved tokens, we can repurpose them to map to the target's special tokens and adapt their semantics through online learning. Also, we could consider incorporating separate embeddings for these special tokens and training them along with the drafter similar to AdaptiVocab (reference [35]), although this might not be optimal for on-device deployment.
>
>
> ### Q3: Strategies for N-gram Cache Optimization on Edge Devices
>
> #### Current Cache Footprint & Growth Rate
>
> - **Clarifications**: Thank you for the question. ``Table 2`` shows the n-gram cache memory footprint across tasks, and ``Table 3`` indicates the growth of the n-gram cache size as training progresses on the MBPP+HumanEval dataset. The growth rate is defined as ``growth_rate = cache_size / training_tokens``
> - **Reasoning**: Overall, n-gram cache size across tasks remains relatively small compared to the size of the draft or target model. This indicates potential feasibility for on-device setting given the small overhead. We can also observe the cache size increases sub-linearly with respect to the amount of training tokens, and should eventually plateau as the growth rate decreases. This is likely the result of the cache capturing most of the high frequency n-grams.
> - Considering the cache overhead is minimal in our selected model pairs, it may not be the bottleneck for edge deployment. That is one of the reasons for our current Python dictionary based n-gram cache implementation. It offers average-case O(1) time complexity for both insertion and lookup, making it highly efficient for real-time access, at the cost of higher memory usage, primarily due to the overhead associated with hash table storage.
>
> Table 2. Final N-gram cache size across tasks with Llama-68M drafter and Qwen2-7B target. The N-gram cache is implemented as a normal python dict.
> *Cache memory (MB) is derived from `pympler.asizeof.asizeof()`.*
>
> ||GSM8K|MBPP+HumanEval|Alpaca|XSUM|
> |-|-|-|-|-|
> |**Training Samples**|7473|910|8000|4000|
> |**Cache Size (#n-grams)**|5569|2238|20339|17013|
> |**Cache Memory (MB)**|1.372|0.501|4.569|3.924|
>
> Table 3. N-gram cache growth rate during online learning for MBPP+HumanEval
>
> |Training Step|100|200|300|400|500|600|700|800|900|
> |-|-|-|-|-|-|-|-|-|-|
> |**Cache Size (#n-grams)**|667|1021|1256|1466|1628|1791|1929|2089|2208|
> |**Training Tokens (#tokens)**|26K|57K|85K|113K|141K|167K|195K|224K|252K|
> |**Cache Growth Rate**|0.0247|0.0177|0.0147|0.0128|0.0114|0.0106|0.0098|0.0093|0.0087|
>
> #### Cache Optimization Strategies
>
> - **Clarifications**: In the case that memory is a limiting factor, one alternative is to use the prefix tree (Trie). Tries are particularly well-suited for n-gram storage because they exploit common prefixes among n-grams, leading to better memory efficiency. While Tries typically have slightly slower insertion and lookup times compared to hash maps, this trade-off can be acceptable when memory is a primary concern. Additionally, compression techniques such as radix trees (compressed tries) can be applied to further reduce memory usage.
> - Beyond structural changes, we can also consider implementing eviction policies to control the size of the cache, either by limiting the number of cached n-grams or by capping total memory usage. Common strategies include: Least Recently Used (LRU) that evicts the n-gram that hasn’t been accessed for the longest time, Least Frequently Used (LFU) that removes the n-gram with the lowest access frequency and hybrid policies that combine recency and frequency to better reflect usage patterns.
> - **Experiments**: `Table 4` indicates how speedup is impacted by different max cache size and cache eviction policies: LRU and LFU. `1/4` and `1/2` refer to using max cache size as a quarter and half of the full final cache size that we previously trained. `No Cache` and `Full Cache` are the same as $L_{DM}$ and $L_{DM}+\lambda L_{N-gram}$ in Table 1 in the original paper.
> - **Reasoning**: Among all eviction policies and max cache sizes, both evaluation metrics are better than using no cache. Specifically LRU shows diminishing returns over the growth of the cache. Meanwhile LFU plateaus at an earlier stage, with performance matching the full cache even with a quarter of the size. This could be due to LFU capturing most of the high frequency and important n-grams, compared to LRU which only captures the recent n-grams. In the context of on-device deployment, this implies that with proper selection of cache size and eviction strategy, we can potentially attain robust performance given only constrained memory resources, highlighting the practical feasibility of our approach.
>
> Table 4. Cache size scaling v.s. Speedups on MBPP+HumanEval with Llama-68M as the drafter and Qwen2-7B as the target model.
>
> |Cache Size|Cache Type|Accept Rate|Speedup|
> |-|-|-|-|
> |No Cache|-|0.219|1.29x|
> |1/4|LRU|0.254|1.33x|
> ||LFU|0.259| 1.36x|
> |1/2|LRU|0.261|1.35x|
> ||LFU|0.263|1.36x|
> |Full Cache|-|0.267|1.36x|
>
>
> ### Q4: Drawbacks with the "dynamic adapter switching" approach (LoRA)
>
> - **Clarifications**: Thank you for bringing up the question. We can identify two immediate challenges arising from the LoRA approach in OmniDraft
>     - Challenge 1: Depending on the hardware and framework used, there could be some overhead with dynamically switching LoRA adapters depending on the logic used. Frequent switching could also impact memory movement and maybe caching systems. It all depends on the final implementation of these frameworks on the hardware.
>     - Challenge 2: Regarding fixed-point quantization, which would become a necessity depending on the edge-device used, the different adapters could induce varying floating-point activation ranges, which could pose challenges for effective quantization. This can be overcome by using adapter-specific quantization parameters and dynamically switching these parameters based on the target or task.
> - Overcoming these practical challenges should bring us closer to a truly universal drafter.
>
>
> ### Formatting Issues
>
> We sincerely thank the reviewers for highlighting the typos and formatting issues. We apologize for these oversights and will address them in the revised version.

---

> > ### Comment · Reviewer_PzLn · 2025-08-06
> > **Post rebuttal comment**
> >
> > Sincerely thank the authors for their clarifications and answering the questions in detail. The empirical details provided do help to get some detailed understanding of what the authors have in mind for various aspects (n-gram cache, distribution shift) to a great extent. However, there are still some statements in the rebuttal which I feel should be better explained to make the work more methodical (e.g. how cache size and overhead scales, more clear rationale for cache eviction policies, how special tokens leverage semantic understanding and does this really work in practice?, etc).
> >
> > Therefore, I prefer to keep my existing "positive" score.

---

> > > ### Author Response · Authors · 2025-08-07
> > > **Additional Clarifications**
> > >
> > > Dear Reviewer
> > >
> > > We sincerely thank you for your thoughtful and constructive feedback, and for acknowledging the clarifications in our initial rebuttal. We are grateful for your positive evaluation.
> > > We understand your remaining reservations and would like to clarify some of these points.
> > > - **N-gram Cache Clarifications**
> > >     - **Cache size scaling with training data**: During training, the cache size grows **sub-linearly** relative to the number of training tokens and eventually stabilizes as the **growth rate diminishes**. Due to character constraints in the rebuttal, we were unable to include a comprehensive table across multiple tasks. `Table 5` below showcases this behavior for an additional task GSM8K as a representative example. (in addition to `Table 3` in the initial rebuttal for task MBPP+HumanEval). This table shows the scaling behavior of the n-gram cache as training data sizes vary.
> > >     - **Cache overhead**: Based on the trend, the cache size memory overhead (from `~0.4MB` to `~1.3MB` for GSM8k during training) is relatively small compared to the overall size of the drafter (`~128MB` with FP16) and target (`~14GB` with FP16) models . Therefore, it is unlikely to pose a bottleneck for edge deployment scenarios.
> > >     - **Cache size scaling with model size**: Additionally, `Table 6` shows the scaling behavior of the cache as either the draft model or the target model size changes (after training on the GSM8K dataset (4k subset) ). It can be observed that the final cache sizes are mostly similar, indicating the difference in tokenizers between the drafter and target and the training data dictate the cache size more than the model size (unless the target model distributions are significantly different).
> > >
> > > Table 5. N-gram cache scaling or growth rate with training data sizes for GSM8K, with Llama-68M as drafter and Qwen2-7B as target.
> > >
> > > |**Training Step**|500|1000|1500|2000|2500|3000|3500|4000|
> > > |-|-|-|-|-|-|-|-|-|
> > > |**Cache Size (#n-grams)**|1097|1717|2263|2692|3078|3430|3734|4009|
> > > |**Training Tokens (#tokens)**|83K|166K|253K|336K|418K|501K|583K|668K|
> > > |**Cache Growth Rate**|0.0130|0.0102|0.0089|0.0080|0.0073|0.0068|0.0064|0.0060|
> > >
> > > Table 6. N-gram cache scaling with different drafter and target model sizes for GSM8K.
> > >
> > > |Cache Memory (MB)|Qwen2.5-7B|Qwen2.5-14B|Qwen2.5-32B|
> > > |-|-|-|-|
> > > |**Llama-68M**|1.0505|1.0611|1.0562|
> > > |**Llama-160M**|1.0477|1.0612|1.054|
> > >
> > > - **Cache Eviction Policies**: Regarding the various cache eviction policies, the rationale behind the choice of LRU and LFU are multi-fold.
> > >     - **Support and effectiveness**: LRU (Least Recently Used) and LFU (Least Frequently Used) are widely adopted due to their balance between simplicity and performance across diverse scenarios. They are well supported and potentially better optimized for actual on-device deployment. Other policies like FIFO or Random Replacement are less suited to our current scenario, and policies with more complex priority scoring (for example, based on each n-gram's entropy or distillation loss) would add too much overhead.
> > >     - **Task-specific relevance**: In our context, LFU effectively **captures high-frequency and important n-grams**, which is particularly beneficial for online learning tailored to a specific data distribution.
> > >     - **Potential for enhancement**: We could explore hybrid strategies such as LRFU, which combines the strengths of LRU and LFU, that might yield better results and serve as an  additional ablation for the final paper.
> > >
> > > - **Special Tokens**
> > >     - **Semantics**: Our method of treating special tokens as regular text tokens provide a straightforward and effective way to integrate them into our framework. For instance, tokens like `<im_start>` which could be a special token for the target model (Qwen2-7B), would be broken down into n-grams like `<`, `im`, `_`, `start`, `>` by the drafter tokenizer (Llama-68M). During training, the model learns to associate these components through the n-gram cache, effectively capturing the semantic meaning of the original special token within the target vocabulary. As such, for a specific context, the drafter learns to associate the n-gram to the corresponding special token.
> > >     - **Practicality**: Regarding the practicality, this approach is particularly relevant in our current setup, where special tokens appear in chat template prompts. The model learns these mappings during training to ensure proper handling and understanding of such tokens.
> > >
> > > We hope with these additional clarifications, the paper can be considered for a higher evaluation score. Thank you once again.

---

### Official Review · Reviewer_F753 · 2025-07-02

**Clarity:** 3
**Significance:** 3
**Originality:** 3
**Rating:** 4
**Confidence:** 4

**Summary:**

This paper proposes OmniDraft, a framework designed to shift the speculative decoding paradigm for on-device applications from a static, model-specific setup to a universal and adaptive one. The central idea is to enable a single, lightweight draft model to not only serve a variety of target models with incompatible vocabularies but also to continuously improve its performance through online learning from user data. To achieve this, OmniDraft introduces two core interacting mechanisms: an online n-gram cache to resolve cross-vocabulary tokenization mismatches on-the-fly and a hybrid online distillation process that continually aligns the draft model with the target model's verified outputs. These components, combined with adaptive drafting techniques for efficiency, are shown to successfully allow a single Llama-68M to accelerate various models like Qwen2-7B and Llama3-8B by up to 1.5-2x.

**Questions:**

•	The paper correctly identifies the N-gram cache's memory footprint as a potential limitation for on-device use. Could the authors provide a brief empirical analysis of the cache's growth rate and final memory size observed during the experiments? This would be very helpful for understanding the practical scalability of the approach.
•	The experiments effectively demonstrate OmniDraft's performance with target models in the 7-8B parameter range. Have the authors conducted any preliminary experiments, or do they have insights on how the framework would perform with significantly larger target models? The alignment difficulty and potential speedup could change substantially at that scale.

**Ethical Concerns:**

["NO or VERY MINOR ethics concerns only"]

**Final Justification:**

After carefully reading the response, I found that most of my concerns have been addressed, and I decided to keep my positive rating.

**Limitations:**

Yes

**Quality:**

3

**Strengths And Weaknesses:**

Strengths:
•	Significance: Addresses a critical and practical problem in LLM deployment: enabling a single draft model to work with any target model.
•	Originality: The "one drafter for all" online adaptive framework is a novel paradigm, using an online n-gram cache for cross-vocabulary translation and hybrid distillation for live adaptation.
•	Quality: Presents a solid empirical evaluation across various incompatible models and tasks, supported by relevant ablation studies.
•	Clarity: The paper is clearly written, with its methodology clearly outlined in pseudocode and its experimental results presented effectively.


Weaknesses:
•	N-gram Cache Scalability: The scalability and memory footprint of the online n-gram cache, critical for on-device use, are not fully analyzed.
•	Online Learning Stability: Online adaptation can be unstable, as shown by training curves (including in the appendix), raising long-term stability concerns.
•	Modest Speedup: The reported speedup (up to 1.5-2x) is valuable but more modest compared to results from other specialized systems.

---

> ### Author Rebuttal · Authors · 2025-07-31
>
> We would like to express our gratitude to the reviewers for their careful review of our paper and thoughtful feedback. We appreciate the opportunity to clarify the contributions, novelty and limitations of our work. We are grateful to reviewer F753 for recognizing our one-drafter-for-all novel paradigm and the methodology.
>
>
> ### Q1: N-gram Cache Footprint, Growth Rate and Scalability
>
>
> #### Memory Size or Footprint
>
> - **Clarifications**: Thank you for the question. ``Table 1`` below indicates the final N-gram cache memory sizes across tasks.
> - **Reasoning**: Overall N-gram cache size across tasks remain relatively small compared to the size of the draft or target model. This indicates potential feasibility for on-device setting given the small overhead. In addition, we are also working towards a more compact and efficient implementation of the N-gram cache specifically for on-device deployment as part of our future work.
>
> Table 1. Final N-gram cache size across tasks with Llama-68M drafter and Qwen2-7B target. The N-gram cache is implemented as a normal python dict.
> *Cache memory (MB) is derived from `pympler.asizeof.asizeof()`.*
>
> ||GSM8K|MBPP+HumanEval|Alpaca|XSUM|
> |-|-|-|-|-|
> |**Training Samples**|7473|910|8000|4000|
> |**Cache Size (#n-grams)**|5569|2238|20339|17013|
> |**Cache Memory (MB)**|1.372|0.501|4.569|3.924|
>
>
> #### N-gram Cache Growth Rate
>
> - **Clarifications**: ``Table 2 & 3`` indicates the growth of the n-gram cache size as training progresses on GSM8K and MBPP+HumanEval datasets. Growth rate is defined as ``growth_rate = cache_size / training_tokens``
> - **Reasoning**: We observe that the cache size increases sub-linearly with respect to the amount of training tokens, and should eventually plateau as the growth rate decreases. This is likely the result of the cache capturing most of the high frequency n-grams.
>
> Table 2. N-gram cache growth rate during online learning for GSM8K
>
> |**Training Step**|500|1000|1500|2000|2500|3000|3500|4000|
> |-|-|-|-|-|-|-|-|-|
> |**Cache Size (#n-grams)**|1097|1717|2263|2692|3078|3430|3734|4009|
> |**Training Tokens (#tokens)**|83K|166K|253K|336K|418K|501K|583K|668K|
> |**Cache Growth Rate**|0.0130|0.0102|0.0089|0.0080|0.0073|0.0068|0.0064|0.0060|
>
>
> Table 3. N-gram cache growth rate during online learning for MBPP+HumanEval
>
> |Training Step|100|200|300|400|500|600|700|800|900|
> |-|-|-|-|-|-|-|-|-|-|
> |**Cache Size (#n-grams)**|667|1021|1256|1466|1628|1791|1929|2089|2208|
> |**Training Tokens (#tokens)**|26K|57K|85K|113K|141K|167K|195K|224K|252K|
> |**Cache Growth Rate**|0.0247|0.0177|0.0147|0.0128|0.0114|0.0106|0.0098|0.0093|0.0087|
>
>
> #### N-gram Cache Scalability
>
> - **Clarifications**: We implemented 2 baseline cache eviction policies with varied max cache size. Due to time constraints, we perform experiments on a single dataset.
> - **Experiments**: `Table 4` indicates how speedup is impacted by different max cache size and cache eviction policies: Least Recently Used (LRU) that evicts the n-gram that hasn’t been accessed for the longest time, Least Frequently Used (LFU) that removes the n-gram with the lowest access frequency. `1/4` and `1/2` refer to using max cache size as a quarter and half of the full final cache size that we previously trained.
> - **Reasoning**: Among all eviction policies and max cache sizes, both evaluation metrics are better than using no cache. Specifically LRU shows diminishing returns over the growth of the cache. Meanwhile LFU plateaus at an earlier stage, with performance matching the full cache even with a quarter of the size. This could be due to LFU capturing most of the high frequency and important n-grams, compared to LRU which only captures the recent n-grams. In the context of on-device deployment, this implies that with proper selection of cache size and eviction strategy, we can potentially attain robust performance given only constrained memory resources, highlighting the practical feasibility of our approach.
>
>
> Table 4. Cache size scaling v.s. Speedups on MBPP+HumanEval
>
> |Cache Size|Cache Type|Accept Rate|Speedup|
> |-|-|-|-|
> |No Cache|-|0.219|1.29x|
> |1/4|LRU|0.254|1.33x|
> ||LFU|0.259| 1.36x|
> |1/2|LRU|0.261|1.35x|
> ||LFU|0.263|1.36x|
> |Full Cache|-|0.267|1.36x|
>
>
>
>
>
> ### Q2: Performance with Larger Models.
>
>
>
> #### Different and Larger LLMs.
> - **Clarification**:  Thank you for the suggestion. As per the request, we performed additional experiments on larger LLMs as shown in `Table 5`.
> - **New Experiments**: We used Qwen2.5 series of models of larger sizes including **7B, 14B and 32B** parameters. Datasets used are GSM8k and MBPP+HumanEval. As an additional ablation, we also performed analysis on a larger drafter of size **160M** parameters. The performance on these larger models are consistent with our previous results, with larger target models resulting in **2x** improvement on the GSM8K dataset.
> - **Larger Drafter**: Due to time and GPU constraints, we were able to perform these ablations on 2 datasets and up-to 32B parameter model only. The Llama-160M parameter model was chosen as the larger drafter after taking into consideration the increased latency of the drafter model compared to Llama-68M. Overall, the larger drafter model does introduce a reduction in the overall speedup across all target models, however, due to its enhanced potential capability, the acceptance rate does improve across all the target models.
> - **Larger Targets**: The Omnidraft framework demonstrates strong scalability as the size of the target LLM increases, as shown in the table. While the gap between the drafter and the target model grows with model size, the drafter remains the limiting factor. During training, the drafter progressively aligns with the target; however, once it reaches its alignment capacity, further increase in target size continues to yield speedup improvements, while the acceptance rate plateaus. For the larger drafter, the acceptance rate does indeed improve, although speedup does take a hit due to the increased latency of the drafter.
>
>
> Table 5. Comparison of Cross-vocabulary Distillation Performance with Llama-68M and Llama-160M across Three Target Models. Due to time and GPU constraints, scaling experiments on the GSM8K dataset are only using a 4K subset.
>
> |Target|Method|GSM8K Acc|GSM8K Speedup|MBPP+HumanEval Acc|MBPP+HumanEval Speedup|
> |-|-|-|-|-|-|
> |Qwen2.5 7B|SpD_UAG-DM (68M)|0.15|1.02x|0.10|0.94x|
> ||*L_DM + λ L_N-gram* (68M)|**0.401**|**1.66x**|**0.27**|**1.33x**|
> ||SpD_UAG-DM (160M)|0.18|0.70x|0.13|0.60x|
> ||*L_DM + λ L_N-gram* (160M)|**0.47**|**1.12x**|**0.32**|**0.90x**|
> |Qwen2.5 14B|SpD_UAG-DM (68M)|0.15|1.17x|0.10|1.14x|
> ||*L_DM + λ L_N-gram* (68M)|**0.407**|**1.92x**|**0.272**|**1.57x**|
> ||SpD_UAG-DM (160M)|0.178|0.89x|0.13|0.84x|
> ||*L_DM + λ L_N-gram* (160M)|**0.472**|**1.40x**|**0.33**|**1.19x**|
> |Qwen2.5 32B|SpD_UAG-DM (68M)|0.153|1.30x|0.10|1.23x|
> ||*L_DM + λ L_N-gram* (68M)|**0.42**|**2.05x**|**0.274**|**1.71x**|
> ||SpD_UAG-DM (160M)|0.187|1.03x|0.133|0.97x|
> ||*L_DM + λ L_N-gram* (160M)|**0.49**|**1.62x**|**0.335**|**1.40x**|
>
>
>
>
> ### Q3: Online Learning Stability and Modest Speedup
>
>
> #### Online Learning
> - **Clarifications**: Regarding online learning, the observed instability is an inherent characteristic of the setting, which occurs as fluctuations (e.g., zig-zag patterns) in the training curves. However, as training progresses and the n-gram cache becomes more populated and captures a broader set of in-distribution data, the instability can be potentially mitigated by training for much larger samples. We can also incorporate other techniques such as replay buffer or other aggressive regularization techniques for online learning instability mitigation.
>
> #### Speedup
> - **Clarifications**: In terms of speedup, we observe a consistent improvement as the target LLM size increases from 7B to 32B. Additionally, our framework could potentially benefit from incorporating advanced SpD techniques such as Medusa (which introduces multiple token prediction heads on the drafter) and various tree-based sampling strategies. However, these methods are orthogonal to our approach and are expected to provide further gains when layered on top of our current system.

---

> > ### Comment · Reviewer_F753 · 2025-08-03
> >
> > Thank you for the feedback. I suppose most of my concerns have been addressed and I am pleased to keep my positive rating.

---

> > > ### Author Response · Authors · 2025-08-03
> > >
> > > Thank you for your thoughtful engagement with our work. We're pleased that our rebuttal addressed your concerns. Your feedback has been valuable in strengthening our paper, and we appreciate your continued positive assessment.

---

> > > > ### Author Response · Authors · 2025-08-07
> > > >
> > > > Dear Reviewer,
> > > >
> > > > Thank you again for your positive assessment last week. You mentioned that most of your concerns were addressed, and as we're still in the discussion period, we were wondering whether there are any more concerns that require clarifications.
> > > > If not, we hope our rebuttal discussions have resolved most of your concerns and that these improvements might merit a stronger positive rating.
> > > >
> > > > Thank you very much.

---

### Official Review · Reviewer_NjJe · 2025-07-03

**Clarity:** 2
**Significance:** 3
**Originality:** 3
**Rating:** 4
**Confidence:** 4

**Summary:**

This paper presents OmniDraft, a speculative-decoding framework that lets a single 68 M-parameter drafter accelerate multiple large-model back-ends (Vicuna-7B, Qwen2-7B, Llama-3-8B) even when their tokenizers differ. Three technical pieces enable this:

1. Cross-vocabulary n-gram cache. During inference, the system merges sequences of draft sub-tokens into target tokens via a cache that is populated on-the-fly, allowing correct acceptance-ratio tests across vocabularies.

2. Online hybrid distillation. A token-level KL term aligns directly mapped tokens, while an NLL term aligns cached n-grams, continually fine-tuning the drafter as it serves requests.

3. Online adaptive drafting. A lightweight head predicts token-acceptance probabilities and shortens or lengthens drafts to maximise speed per wall-time budget.

**Questions:**

1. End-to-end latency & energy. Please supply absolute wall-time and energy figures for each target on a mobile GPU or CPU to confirm that 1.6–2 × speed-up translates into real user-perceived gains.

2. Cache footprint & scaling. How large does the n-gram cache grow after, say, 10 k prompts, and what is its lookup/update time under concurrency? A memory/throughput chart would clarify on-device feasibility.

3. Robustness to target drift. If the cloud back-end swaps from Qwen2-7B to Qwen2-14B mid-session, how quickly does the drafter re-align, and does the cache hurt or help during the transition?

Addressing (1), (2), (3) with concrete measurements would raise Quality and Significance and push the paper into clear-accept territory.

**Ethical Concerns:**

["NO or VERY MINOR ethics concerns only"]

**Final Justification:**

The clarifications around cache scaling, target drift, and preliminary latency measurements — particularly on mobile hardware — are appreciated and address key concerns.

While some results are still early-stage or pending (e.g., concurrent cache access and full on-device deployment), I find the methodology sound and the direction promising. Given the additional evidence, I am raising my score from 3 to 4, though not strongly. Stronger comparisons and more mature system profiling would further strengthen the case.

**Limitations:**

yes

**Quality:**

2

**Strengths And Weaknesses:**

1. Quality – The empirical study spans four tasks and three heterogeneous targets, with clear ablations on cache utility, loss choices and drafting heads. However, all timing figures are relative; absolute wall-clock latency, GPU utilisation and memory footprint of the cache are absent. Online adaptation is evaluated for only a single epoch and on 68 M→7 B/8 B scales; it remains unclear whether gains persist for larger drafters or longer streams.

2. Clarity – The methodology is carefully derived and Algorithm 1 is explicit. Yet crucial implementation numbers (cache size limits, eviction policy, adaptive-head thresholds) are scattered or deferred to the appendix, and the main text exceeds typical concision norms.
Significance – Decoupling drafter and target tokenizers is an important step toward real on-device speculative decoding. Still, reported speed-ups hover around 1.6 ×; rival approaches such as UAG or AdaptiVocab are mentioned but not compared directly, leaving the incremental benefit uncertain.

3. Originality – The live n-gram cache that learns cross-tokenizer merges during inference is novel, and combining it with continuous distillation and adaptive draft length is an elegant unification.

---

> ### Author Rebuttal · Authors · 2025-07-31
>
> We would like to express our gratitude to the reviewers for their careful review of our paper and constructive feedback. We appreciate the opportunity to clarify the contributions, novelty and limitations of our work. We are grateful to Reviewer NjJe for recognizing the technicality of our methodology, novelty, algorithm and acknowledging the various empirical ablations done.
>
> ### Q1: End-to-end latency on mobile GPU or CPU
>
> - **Clarifications**: Thank you for the suggestion. Regarding the system's end-to-end latency, we plan to implement the entire pipeline on-device as part of our future work. Due to current time constraints, we have profiled both the drafter and target models on both CPU (AMD Ryzen Threadripper PRO 5975WX) and mobile GPU (Apple M4 Pro) using llama-bench from llama.cpp with the best setting we figured out, and then computed the overall speedup based on the offline recorded steps of the SpD process for each proposal in the evaluation set. Please note that this speedup does not account for the ngram-cache lookup, as its on-device implementation is still pending.
>
> - **Experiments**: As shown in `Table 1`, the results on the CPU are promising, demonstrating a potential speedup of approximately 2.5×. On the mobile GPU, however, the performance is slightly less favorable. This is primarily due to the latency of the target model on the mobile GPU, where the AR4 (latency of a verification step with K=4) is ~ 2 * AR1 (latency of a generation step). This is not exactly aligned with speculative decoding's assumption, resulting in reduced overall speedup. This could also potentially be due to an optimal setting that we might have missed in llama-bench for the mobile GPU.
>
> Table 1: Wall-clock latency for drafter and target models on CPU and mobile GPU and the speedup on different datasets
> |Platform|Model|Wall-clock Latency (AR1/AR4, in ms)|GSM8K Speedup|MBPP+HumanEval Speedup|
> |-|-|-|-|-|
> |CPU|Llama-68M|0.37/-|-|-|
> ||Qwen2.5 7B|58.57/59.13|2.47x|1.99x|
> ||Qwen2.5 14B|115.71/116.29|2.50x|2.00x|
> ||Qwen2.5 32B|255.41/255.65|2.61x|2.04x|
> |Mobile GPU|Llama-68M|0.62/-|-|-|
> ||Qwen2.5 7B|34.39/46.18|1.81x|1.45x|
> ||Qwen2.5 32B|149.46/268.06|1.45x|1.13x|
>
> ### Q2: Cache Footprint, Scaling & Growth Rate
>
> - **Clarifications**: Thank you for pointing this out. Our current n-gram cache is implemented using a Python dictionary (hash table-based). This approach provides average-case O(1) time complexity for both insertion and lookup, which ensures high throughput for real-time insertion and lookup. However, our current implementation is not thread-safe for concurrent writes, and thus we are not able to provide the latency with concurrency, which requires an implementation with read-write lock to allow concurrent lookups and serializing writes. And we expect the insertion latency will increase due to the locking mechanism. We'll be looking into that in our on-device implementation.
>
> - **Reasoning**: `Table 2` shows the overall N-gram cache size across tasks remains relatively small compared to the size of the draft or target model. This indicates potential feasibility for on-device setting given the small overhead. In addition, we are also working towards a more compact and efficient implementation of the N-gram cache specifically for on-device deployment as part of our future work.
>
> Table 2. Final N-gram cache size
> *Cache memory (MB) is derived from `pympler.asizeof.asizeof()`.*
>
> ||GSM8K|MBPP+HumanEval|Alpaca|XSUM|
> |-|-|-|-|-|
> |**Training Samples**|7473|910|8000|4000|
> |**Cache Size (#n-grams)**|5569|2238|20339|17013|
> |**Cache Memory (MB)**|1.372|0.501|4.569|3.924|
>
> - **Reasoning**: `Table 3` indicates the growth of the n-gram cache size as training progresses. Growth rate is defined as `growth_rate = cache_size / training_tokens`. We can observe the cache size increases sub-linearly with respect to the amount of training tokens, and should eventually plateau as the growth rate decreases. This is likely the result of the cache capturing most of the high frequency n-grams. We tested various eviction policies across cache sizes and observed diminishing returns, suggesting cache memory can be further reduced with minimal performance loss.
>
> Table 3. N-gram cache growth rate for MBPP+HumanEval
>
> |Training Step|100|200|300|400|500|600|700|800|900|
> |-|-|-|-|-|-|-|-|-|-|
> |**Cache Size (#n-grams)**|667|1021|1256|1466|1628|1791|1929|2089|2208|
> |**Training Tokens (#tokens)**|26K|57K|85K|113K|141K|167K|195K|224K|252K|
> |**Cache Growth Rate**|0.0247|0.0177|0.0147|0.0128|0.0114|0.0106|0.0098|0.0093|0.0087|
>
> ### Q3: Robustness to Target Drift
> #### Target Model Drift
> - **Clarification**:Thank you for the suggestion. We conducted empirical analysis on model switching between Qwen2.5 7B and 14B at arbitrary step intervals (e.g., 7B → 14B → 7B)
> - **Experiments 1**: As shown in `Table 4`, our observations indicate that the drafter model is able to re-align seamlessly after each switch. Notably, switching to the 14B model yields improved speedup due to its higher latency, while maintaining alignment quality. This behavior is largely attributed to the fact that the target models belong to the same model family and as such the drafter is able to continually align during training.
> - **Experiments 2**: We also evaluated the impact of enabling the n-gram cache during target switching. Results show that the drafter recovers alignment quickly when the cache is enabled. However, in scenarios involving cross-family model swaps, a new n-gram cache is preferred due to differences in tokenization. In such cases, a simple distribution shift is insufficient to maintain alignment.
> Since we are unable to include plots directly, we have provided below a tabular summary of the key results to convey the main trends. The corresponding plots will be included in the final version of the paper.
>
> Table 4. Summary of Speedup Transitions Across Training Steps and Model Switches (7B → 14B → 7B)
>
> |Step Range|Target Model|Speedup Trend|Speedup Value|Notes|
> |-|-|-|-|-|
> |0–300|7B|Gradual increase|~0.8 → ~1.05|Initial training phase|
> |300 (Switch)|→ 14B|Immediate jump|~1.4|Transition to larger model|
> |300–600|14B|Slight increase|~1.45|System adapts to new model|
> |600 (Switch)|→ 7B|Drop in speedup|~1.15|Reverting to original model|
> |600–1000|7B|Slight increase|~1.2|Final phase with stable performance|
>
> ### Additional Responses
> #### Larger drafter and target scalability
> - **New Experiments**: Due to time and GPU constraints, we were able to perform these ablations on 2 datasets and up-to 32B parameter model only. We used Qwen2.5 series of models with 7B, 14B and 32B parameters. Datasets used are GSM8k and MBPP+HumanEval. As an additional ablation, we also performed analysis on a larger drafter of size 160M parameters (`Table 5`). The performance on these larger models are consistent with our previous results, resulting in 2x improvement on the GSM8K dataset.
> - **Larger Drafter**: The Llama-160M parameter model was chosen as the larger drafter after taking into consideration the increased latency of the drafter model compared to Llama-68M. Overall, the larger drafter model does introduce a reduction in the overall speedup across all target models. However, due to its enhanced potential capability, the acceptance rate does improve across all the target models.
> - **Larger Targets**: On scalability, as the gap between the drafter and target model grows with model size, the drafter remains the limiting factor. During training, the drafter progressively aligns with the target; however, once it reaches its alignment capacity, further increase in target size continues to yield speedup improvements, while the acceptance rate plateaus. For the larger drafter, the acceptance rate does indeed improve, although speedup does take a hit due to the increased latency of the drafter.
>
> Table 5. Performance with Large Models.
> Due to time and GPU constraints, scaling experiments on the GSM8K dataset are only using a 4K subset.
>
> |Target|Method|GSM8K Acc|GSM8K Speedup|MBPP+HumanEval Acc|MBPP+HumanEval Speedup|
> |-|-|-|-|-|-|
> |Qwen2.5 7B|SpD_UAG-DM (68M)|0.15|1.02x|0.10|0.94x|
> ||*L_DM + λ L_N-gram* (68M)|**0.401**|**1.66x**|**0.27**|**1.33x**|
> ||SpD_UAG-DM (160M)|0.18|0.70x|0.13|0.60x|
> ||*L_DM + λ L_N-gram* (160M)|**0.47**|**1.12x**|**0.32**|**0.90x**|
> |Qwen2.5 14B|SpD_UAG-DM (68M)|0.15|1.17x|0.10|1.14x|
> ||*L_DM + λ L_N-gram* (68M)|**0.407**|**1.92x**|**0.272**|**1.57x**|
> ||SpD_UAG-DM (160M)|0.178|0.89x|0.13|0.84x|
> ||*L_DM + λ L_N-gram* (160M)|**0.472**|**1.40x**|**0.33**|**1.19x**|
> |Qwen2.5 32B|SpD_UAG-DM (68M)|0.153|1.30x|0.10|1.23x|
> ||*L_DM + λ L_N-gram* (68M)|**0.42**|**2.05x**|**0.274**|**1.71x**|
> ||SpD_UAG-DM (160M)|0.187|1.03x|0.133|0.97x|
> ||*L_DM + λ L_N-gram* (160M)|**0.49**|**1.62x**|**0.335**|**1.40x**|
>
> #### Online learning and Speedup numbers
> - **Clarification**:  Speedup metrics were measured using absolute wall-clock latency on a single 40GB/80GB NVIDIA A100 GPU, averaged over three independent runs on the evaluation set. Due to the online nature of our setting, training was limited to a single epoch or a subset of samples. However, we performed experiments across four different tasks, each with varying sample sizes, providing a broader perspective on the observed speedup trends.. We observe that as we increase the target models, we get a larger speedup of  ~2x. We expect even larger speedup if we use the 70B target model.
>
> #### Comparisons to baseline
> - **Clarification**:
> Our baseline comparison for SpD-DM is equivalent to the UAG method. On all our results, our approach consistently outperforms the UAG baseline.
> Regarding the AdaptiVocab method, it is less suitable for on-device deployment, as it requires modifying the embedding matrix by adding new token embeddings i.e. effectively altering the architecture of the draft model. Other techniques, such as Medusa, are orthogonal to our setting and could be integrated on top of our framework to further enhance performance.

---

### Official Review · Reviewer_8Gpq · 2025-07-04

**Clarity:** 3
**Significance:** 3
**Originality:** 2
**Rating:** 5
**Confidence:** 3

**Summary:**

This paper introduces OmniDraft, a unified speculative decoding framework that allows a single, efficient draft model to work seamlessly with any target language model, even in online and on-device settings. The approach combines an online n-gram cache and hybrid distillation fine-tuning to resolve vocabulary mismatches between draft and target models, and uses adaptive drafting techniques to further accelerate decoding. The authors demonstrate that OmniDraft can pair a single small draft model with multiple large target models across diverse tasks achieving notable speedups of up to 2x without sacrificing flexibility or performance.

**Questions:**

1. In terms of scalability, could you share any intuition of how the performance of this new speculative decoding method changes as the size of LLMs gets larger?

**Ethical Concerns:**

["NO or VERY MINOR ethics concerns only"]

**Final Justification:**

I raise my score to 5. The authors addressed my comments well with additional experiments.

**Quality:**

3

**Strengths And Weaknesses:**

Strength

s1: This paper is well written and easy to follow

s2: This paper comes up with the novel idea of solving incompatibility issue between draft and target model by integrating an online n-gram cache

s3: This paper experimentally shows that their method achieves better performance and speed ups compared to previous baselines.

Weakness

w1: I recommend authors to do more experiments with more downstream tasks and different/larger LLMs

w2: It would be better if authors can add more baselines other than SpD.

---

> ### Author Rebuttal · Authors · 2025-07-30
>
> We would like to express our gratitude to the reviewers for their careful review of our paper and constructive feedback. We appreciate the opportunity to clarify the contributions, novelty and limitations of our work. We are grateful to Reviewer 8Gpq for recognizing the novelty of our method and acknowledging the strength of our experimental results.
>
>
> ### Q1: I recommend authors to do more experiments with more downstream tasks and different/larger LLMs; in terms of scalability, could you share any intuition of how the performance of this new speculative decoding method changes as the size of LLMs gets larger?
>
>
> #### Different and Larger LLMs
>
> - **Clarifications**: Thank you for the question. As per the request, we performed additional experiments on larger LLMs.
> - **Experiments**: We used Qwen2.5 series of models of larger sizes including **7B, 14B and 32B** parameters. Datasets used are GSM8k and MBPP+HumanEval. As an additional ablation, we also performed analysis on a larger drafter of size **160M** parameters. The performance on these larger models are consistent with our previous results, with larger target models resulting in **2x** improvement on the GSM8K dataset.
> - **Reasoning**: Due to time and GPU constraints, we were able to perform these ablations on 2 datasets and only up-to 32B parameter model. The Llama-160M parameter model was chosen as the larger drafter after taking into consideration the increased latency of the drafter model compared to Llama-68M. Overall, the larger drafter model does introduce a reduction in the speedup across all target models, however, due to its enhanced potential capability, the acceptance rate does improve across all the target models. Due to time constraints, we are unable to perform further experiments on more downstream tasks.
>
>
> #### Scalability
>
> - **Clarification**: The Omnidraft framework demonstrates strong scalability as the size of the target LLM increases, as shown in the `Table 1` below.
> - **Reasoning**: The key intuition is that while the gap between the drafter and the target model grows with model size, the drafter remains the limiting factor. During training, the drafter progressively aligns with the target; however, once it reaches its alignment capacity, further increase in target size continues to yield speedup improvements, while the acceptance rate plateaus. As such, when we increase the drafters capacity to 160m, the acceptance rate does indeed improve, although speedup does take a hit due to the increased latency of the drafter. Due to time and hardware constraints, we were unable to conduct experiments with the Qwen2.5 72B target model.
>
> Table 1. Performance with Large Models. Due to time and GPU constraints, scaling experiments on the GSM8K dataset are only using a 4K subset.
>
> | Target        | Method                                          | GSM8K Acc | GSM8K Speedup | MBPP+HumanEval Acc | MBPP+HumanEval Speedup |
> |---------------|-------------------------------------------------|-----------|----------------|---------------------|--------------------------|
> | Qwen2.5 7B    | SpD_UAG-DM (68M)                             | 0.15      | 1.02x          | 0.10                | 0.94x                    |
> |               | *L_DM + λ L_N-gram* (68M)                   | **0.401** | **1.66x**      | **0.27**            | **1.33x**                |
> |               | SpD_UAG-DM (160M)                            | 0.18      | 0.70x          | 0.13                | 0.60x                    |
> |               | *L_DM + λ L_N-gram* (160M)                  | **0.47**  | **1.12x**      | **0.32**            | **0.90x**                |
> | Qwen2.5 14B   | SpD_UAG-DM (68M)                             | 0.15      | 1.17x          | 0.10                | 1.14x                    |
> |               | *L_DM + λ L_N-gram* (68M)                   | **0.407** | **1.92x**      | **0.272**           | **1.57x**                |
> |               | SpD_UAG-DM (160M)                            | 0.178     | 0.89x          | 0.13                | 0.84x                    |
> |               | *L_DM + λ L_N-gram* (160M)                  | **0.472** | **1.40x**      | **0.33**            | **1.19x**                |
> | Qwen2.5 32B   | SpD_UAG-DM (68M)                             | 0.153     | 1.30x          | 0.10                | 1.23x                    |
> |               | *L_DM + λ L_N-gram* (68M)                   | **0.42**  | **2.05x**      | **0.274**           | **1.71x**                |
> |               | SpD_UAG-DM (160M)                            | 0.187     | 1.03x          | 0.133               | 0.97x                    |
> |               | *L_DM + λ L_N-gram* (160M)                  | **0.49**  | **1.62x**      | **0.335**           | **1.40x**                |
>
>
> ### Q2: It would be better if authors can add more baselines other than SpD.
>
> #### Additional Baselines
> - **Clarifications**: Thank you for the suggestion. Our current baseline mentioned in all our results is SpD_DM, which is our implementation of SpD with direct mapping tokens only and is equivalent to UAG.
> - **Reasoning**: Baselines such as Medusa are orthogonal to our approach and can be integrated on top of our current framework, like multi-head decoding, tree-based attention, and sparsification (SWIFT). In contrast, techniques like EAGLE rely on a shared language model head, which assumes a consistent model architecture and vocabulary. This dependency makes EAGLE unsuitable for our cross-vocabulary setting.
> As there are only a few existing methods that explicitly address cross-vocabulary scenarios, we primarily compare against the UAG approach, which is most relevant to our setup.
> - **References**: UAG - reference [45], Medusa - reference [8], SWIFT - reference [48], EAGLE - reference [27] from the main paper.

---

### Note · Authors · 2025-08-11

We sincerely thank the AC and all reviewers for their detailed and constructive feedback. We especially appreciate the reviewers' recognition of the novelty of our work in making cross-vocabulary support feasible and enhancing the speculative decoding algorithm with our n-gram cache.

Our key novelty includes:
- Enabling cross-vocabulary speculative decoding through **online n-gram cache**.
- **Hybrid online distillation** with a dedicated n-gram loss function for improved alignment.
- Joint training of online alignment and **adaptive drafting**  for dynamic draft length.

Incorporating our various strategies provides an overall speedup of 1.5-2x across various target model sizes with a single draft model, showcasing the feasibility of *one-drafter-for-all* paradigm for on-device speculative-decoding.

Key rebuttal clarifications:
- **Resolved all major concerns**: We confirm that we have resolved the major points raised by all of the reviewers and have been acknowledged by them. Our revision now includes extensive ablation and results on **larger target (Qwen 2.5 7B/14B/32B) and draft models (Llama 68M/160M)**, and comprehensive ablations on **cache scalability and eviction policies (LRU/LFU)**.
- **Demonstrated robustness and stability**: We performed new ablations on both **model and data distribution** shifts as requested by the reviewers. These results show our n-gram cache provides **improved stability** across all scenarios, a key strength of our approach and also helps **stabilize online learning**.
- **On-Device feasibility**: To demonstrate practical applicability, we included initial performance results on actual on-device hardware, highlighting the potential for considerable speedup. As part of our forward-looking vision, we are also developing a full-fledged end-to-end pipeline that includes optimizations for complex real-world scenarios.

We believe these extensive additions make the paper substantially more rigorous, coherent, and practically grounded. We are confident the revised manuscript has **addressed all critical concerns** and hope it will be considered favorably.

---

### Decision · Program_Chairs · 2025-09-17

**Decision:**

Accept (poster)

**Comment:**

This paper proposes OmniDraft, a unified framework enabling a single draft model to work with any target model and adapt dynamically to user data for on-device speculative decoding. Some reviewers expressed positive comments highlighting its novelty and practical contributions, while others raised concerns about aspects like cache scalability and online learning stability. After discussion, the authors addressed major concerns through additional experiments, including larger model evaluations, cache scaling analysis, and robustness checks on distribution shifts.

Therefore, the AC recommends to accept this paper.